# The GERB Obs4MIPs: a dataset for evaluating diurnal and monthly variation in top of atmosphere radiative fluxes in climate models

Jacqueline E. Russell[1], Richard J. Bantges[1,2], Helen E. Brindley[1,2], Alejandro Bodas-Salcedo[3]

[1]Physics Department, Imperial College, London, SW7 2BX, UK
[2]NERC National Centre for Earth Observation, Imperial College, London, SW7 2BX, UK
[3]Met Office Hadley Centre, Exeter, EX1 3PB, UK

*Correspondence to*: Jacqueline E. Russell (j.e.russell@imperial.ac.uk)

**Abstract.** A newly available radiative flux dataset, specifically designed to enable the evaluation of the diurnal cycle in top of the atmosphere fluxes, as captured by climate and Earth-system models is presented. Observations over the period 2007-2012
made by the Geostationary Earth Radiation Budget (GERB) instrument are used to derive monthly hourly mean outgoing longwave (OLR) and reflected shortwave (RSW) fluxes on a regular 1 degree latitude/longitude grid approximately covering 60° N-60° S and 60° E-60° W. The impact of missing data is evaluated in detail, and a data-filling solution is implemented using estimates of the broadband fluxes from the Spinning Enhanced Visible and Infrared Imager, flying on the same Meteosat platform, scaled to the GERB observations. This relatively simple approach is shown to deliver an approximate factor of ten
improvement in both the bias caused by missing data and the associated variability in the error. To demonstrate the utility of this V1.1 filled GERB 'Obs4MIPs' dataset, comparisons are made to radiative fluxes from two climate configurations of the Hadley Centre Global Environmental model: HadGEM3-GC3.1 and HadGEM3-GC5.0. Focusing on marine stratocumulus and deep convective cloud regimes, diurnally resolved comparisons between the model and observations highlight discrepancies between the model configurations in terms of their ability to capture the diurnal amplitude and phase of the top
of atmosphere fluxes, details that cannot be diagnosed by comparisons at lower temporal resolution. For these cloud regimes the GC5.0 configuration shows improved fidelity with the observations relative to GC3.1 although notable differences remain. The V1.1 filled GERB Obs4MIPs monthly hourly TOA fluxes are available from the Centre for Environmental Data Analysis with the OLR fluxes accessible at https://doi.org/10.5285/90148d9b1f1c40f1ac40152957e25467 (Bantges et al. 2023a) and the RSW fluxes at https://doi.org/10.5285/57821b58804945deaf4cdde278563ec2 (Bantges et al. 2023b).

## 25 1 Introduction

The balance between the Earth's incoming and outgoing radiant energy at the top of the atmosphere, known as the Earth's Radiation Budget (ERB) is the primary driver of the climate system. This Essential Climate Variable is hence a fundamental quantity for understanding the Earth's climate and its variability. Satellite measurements of the earth reflected shortwave (reflected solar) and emitted thermal infrared (outgoing longwave) components of the ERB with dedicated broadband
instruments began in 1975 with the ERB instrument on Nimbus 6 (Smith et al. 1977). Global observations spanning many

years have been obtained from low earth orbit satellites by the Earth Radiation Budget Experiment (ERBE) (Barkstrom, 1984) and Clouds and the Earth's Radiant Energy System (CERES) instruments (Wielicki et al., 1996), and for the tropics by the Scanner for Radiation Budget (ScaRaB) instrument on Megha-Tropiques (Roca et al., 2015). However, the Geostationary Earth Radiation Budget (GERB) experiment (Harries et al., 2005) is the only ERB mission to fly in geostationary orbit and thus the only mission to provide high time resolution broadband observations of the top of atmosphere (TOA) energy.

Four GERB instruments have been deployed sequentially on the four Meteosat Second Generation satellites (Meteosat-8, 9 10 and 11). Since May 2004 they have provided TOA outgoing longwave (OLR) and reflected solar (RSW) flux products broadly covering the geographical region 60º E to 60º W and 60º N to 60º S at a 15-minute temporal resolution. The frequency and longevity of the observations enables the diurnal cycle to be resolved and facilitates the study of fast climate processes, such as cloud and aerosol, by quantifying their changing effect on the radiation balance over a range of timescales from minutes to years. Although the GERB data are only available for the portion of the globe observable from the Meteosat geostationary orbit they provide broadband observations throughout the diurnal cycle. In contrast other temporally resolved radiation budget datasets, such as the CERES Synoptic (SYN) products, use narrow band geostationary imager data to provide temporal resolution, which supplement, and are scaled to, the much lower temporal resolution broadband observations from the low earth orbiting CERES instruments themselves. GERB data have been used in the development and evaluation of the CERES temporal interpolation used within the SYN products (Doelling et al, 2013, 2016). The instantaneous GERB products have also been used to study and characterise diurnal variability (e.g. Comer et al. 2007, Gristey et al. 2018), the effects of cloud and aerosol on the radiation budget (e.g. Futyan et al. 2005, Slingo et al. 2006, Brindley and Russell 2009, Pearson et al. 2010, Ansell et al. 2014, Banks et al. 2014) and to evaluate the representation of these processes in selected numerical weather prediction and climate models (e.g. Allan et al. 2007, 2011, Greuell et al. 2011, Haywood et al. 2011, Mackie et al. 2017).

While the instantaneous GERB data have been extensively exploited, they are not currently provided in a format that facilitates easy comparison with climate or Earth-system model output. In particular, they suffer from irregular spatial sampling, have a temporal resolution that is higher than that at which model radiation outputs are typically retained, and have a non-standard data format. This paper describes the production of a new monthly hourly mean data product, derived from the instantaneous GERB data, to circumvent these issues. This GERB 'Obs4MIPs' dataset consists of monthly hourly mean TOA OLR and RSW fluxes provided at a 1 degree longitude/latitude spatial resolution for the GERB observation region. It provides a record covering several years that resolves the diurnal variation in the TOA OLR and RSW and is compatible with climate model output such as that produced for the recent Coupled Model Intercomparison Project 6 (CMIP6) (Eyring et al. 2016). The data are provided in Climate and Forecast (CF) v1.7 compliant netCDF format meeting the Observations for Climate Model Intercomparison Projects (Obs4MIPs) submission requirements (Waliser et al. 2020). In the following sections we outline the methodology and provide a detailed analysis of the impact of missing data. We propose and evaluate a relatively simple approach to fill data gaps before providing an illustration of how the new dataset may be employed to assess climate model performance.

## 2 Production of the GERB Obs4MIPs monthly hourly average products

Two versions of the GERB Obs4MIPs monthly hourly average products have been released. The first version (GERB-HR-ED01-1-0) (Bantges et al. 2021a and Bantges et al. 2021b) is produced solely from the GERB data that are available, hereafter referred to as V 1.0 or 'unfilled' GERB Obs4MIPs products. The second improved version (GERB-HR-ED01-1-1) (Bantges et al. 2023a and Bantges et al. 2023b), which is the primary focus of this paper, uses supplementary information derived from the narrow-band Spinning Enhanced Visible and Infrared Imager (SEVIRI) flying on the same Meteosat $2^{nd}$ Generation

platform as GERB (Schmetz et al. 2002) scaled to GERB to fill missing hours of GERB data before calculating the monthly hourly average. These products are referred to hereafter as V 1.1 or 'filled' GERB Obs4MIPs products: we show how they are an improvement over the V 1.0 release in both the amount of data available and the associated uncertainty.

### 2.1 Baseline Methodology

The GERB Obs4MIPs and OLR and RSW fluxes discussed here are based on the observational record from the GERB-1

instrument on Meteosat-9, which runs from May 2007 to January 2013. As noted above, the goal is to create monthly mean, diurnally resolved OLR and RSW fluxes at hourly resolution on a regular 1 degree latitude/longitude grid.

The starting point for creating the averages are the GERB level 2 High Resolution (HR) flux products (Brindley and Russell, 2017) which are produced to facilitate averaging and re-gridding of the GERB instantaneous fluxes. The GERB HR fluxes are a temporally interpolated, resolution enhanced version of original GERB observations derived using spatial information on the

scene variation within the GERB footprint from the SEVIRI instrument. GERB HR fluxes are presented on a regular viewing angle grid which has a spatial resolution of 9 km at the sub-satellite point. They give a 'snapshot' of the fluxes at a 15-minute temporal resolution, aligned to the observation times of the SEVIRI instrument flying on the same satellite.

The GERB instrument operates with the use of a rotating mirror which effectively steps the linear detector array, aligned approximately north-south with respect to the Earth, from east to west and then west-east across the Earth's disc. Early in the

mission the mirror briefly became stuck in a position which allowed direct solar illumination of a portion of the detector array resulting in several pixels being lost. To circumvent the possibility of this reoccurring, subsequent operations were restricted such that diurnally resolved observations are not collected for around 5 weeks either side of the equinoxes. As a result, the production of unfilled GERB Obs4MIPs monthly hourly fluxes was initially restricted to the months of November, December, January and May, June, July, avoiding the months impacted by these operating restrictions. As will be demonstrated in Sect.

2.3 and Sect. 3.2, implementing a relatively simple data filling approach additionally allows the construction of February and August monthly hourly averages within tolerable uncertainties.

Figure 1 summarizes the steps used to produce an unfilled Obs4MIPs product from the GERB HR 15-minute fluxes for both the OLR and RSW. The initial step involves averaging the GERB HR data to an hourly 1 degree latitude/longitude scale. To achieve this, an area weighted average of all the available points whose centres fall within each 1 degree latitude/longitude

grid-box is performed across the region from 60° N-60° S and 60°E-60° W for points with a viewing zenith angle of less than

70°, which is the maximum view angle recommended in the GERB quality summary (Russell, 2017) for averaging to Earth grids. This is followed by a straight average over all the available 15-minute products for each UTC hour, centred on the half hour. When there are no missing data the hourly average of each 1 degree latitude/longitude grid-box would, depending on location, comprise between 6 and 169 GERB HR points at each of the four timeslots obtained during the hour. However, an average is still formed if some timeslots or contributing pixels are missing, as long as there is at least one GERB HR pixel within the 1 degree latitude/longitude grid-box at one time slot in the hourly bin. For the OLR this process is performed directly on the fluxes. For the RSW, the fluxes are converted to albedo before both the spatial and temporal averaging and converted back to flux at the hourly 1 degree latitude/longitude scale, using the incoming solar flux representative of the centre of each 1 degree latitude/longitude grid-box and hourly bin (i.e. at 00:30, 01:30, UTC etc.). As the total solar irradiance and the Earth-Sun distance do not change during the conversion to albedo, and back to flux, this becomes purely an adjustment in solar zenith angle to the centre of the grid box and hour bin. The process is equivalent to multiplying each flux by the ratio $\cos(\theta_{local})$ / $\cos(\theta_{centre})$, where $\theta_{local}$ is solar zenith angle at the HR pixel time and position and $\theta_{centre}$ is the solar zenith angle at the 1 degree latitude/longitude centre at half past the hour. This treatment mitigates for any bias that might result from only some of the 15-minute timeslots within the hour being available and enables hourly fluxes to be derived in the presence of missing data. It also corrects for the variation in solar zenith angle that occurs due to the row to row time variation of the GERB HR which is a consequence of these products being interpolated to match the 12 minute SEVIRI scanning cycle. We note that the GERB HR RSW products use a fixed location independent twilight model based on the model derived from CERES observations (Kato and Loeb, 2003) for solar zenith angles between 85° to 100° and set RSW to zero for solar zenith angles greater than 100°. For consistency this treatment is also applied to the GERB Obs4MIPs products at the daily hourly 1 degree latitude/longitude scale using the solar zenith angle of the centre of the grid-box and hourly bin. Hence, these model twilight and night-time RSW HR fluxes, which are not GERB observations, are not included in the spatial or temporal averaging to the daily hourly 1 degree latitude/longitude scale if the central solar zenith angle is less than 85° but are used to replace grid-box values when the central solar zenith angle is equal to or exceeds 85°. For both OLR and RSW, in the initial 'unfilled' product version, the resulting 1 degree latitude/longitude hourly fluxes are then averaged over all available days of the month to give the final 1 degree latitude/longitude unfilled monthly hourly products.

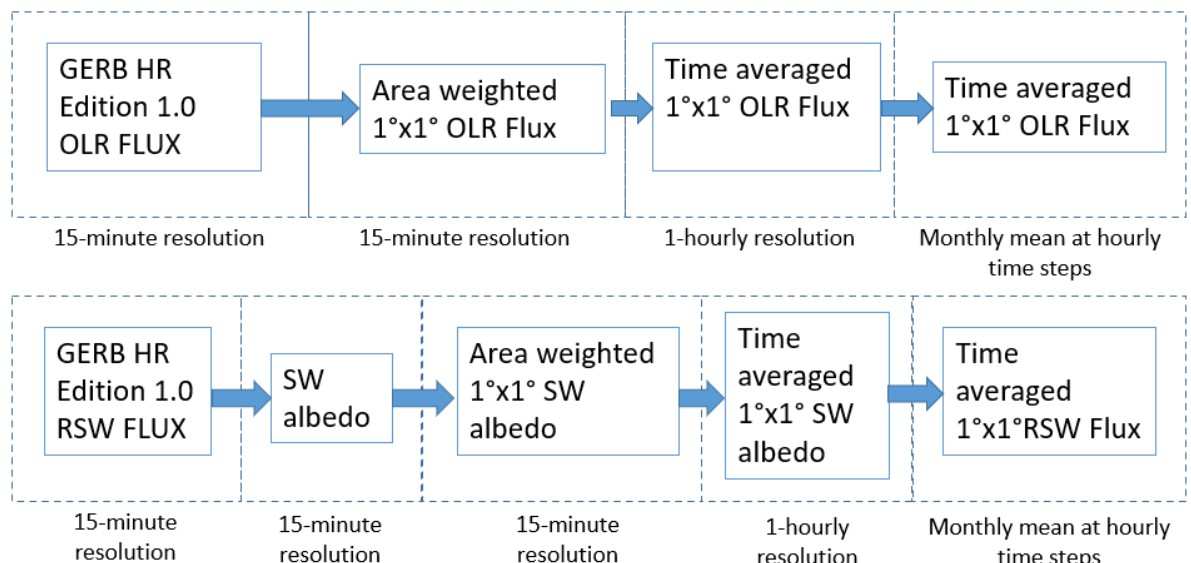

**Figure 1: Schematic of the steps employed in the production of the OLR (top row) and RSW (bottom row) V 1.0 unfilled monthly hourly average Obs4MIPs products from the GERB HR Edition 1 fluxes.**

### 2.2 Missing GERB observations

Calibration operations and other planned and unplanned operational issues result in observational gaps over the whole of the GERB region for one or more hours or, more occasionally, days at a time, and manifest as missing timeslots in the HR record. This leads to a significant number of cases where there are no observations available for a given hour on a particular day, which without further data processing, appear as gaps at the daily hourly scale that result in errors in the Obs4MIPs monthly hourly averages. A summary of the number of missing days of hourly GERB data for the whole GERB-1 record is shown in

Fig. 2 as a function of hour and month. Hours with complete data are shown in white and those with more than 22 days missing are shaded grey. Hours where there are between 1 and 5 missing days in the month are shaded turquoise and cases with between 5 and 22 missing days are shaded pale green. The 5 and 22 missing days boundaries are highlighted as these limits correspond to the maximum number of missing days allowed in the data released for the unfilled and filled products respectively (see Sect. 3). There is an uneven distribution of missing data through the record, with a few months (e.g., December 2012) showing almost complete data coverage and others showing varying degrees of incomplete coverage at all hours. As previously

discussed, operating restrictions in the months around the equinoxes are responsible for an almost complete absence of observations during March, April, September and October resulting in these months being greyed out. These restrictions are also responsible for the pattern of missing data in February and August, where the latter part of these months is always missing. Persistently higher amounts of missing data in the hours around midnight for November and May is a result of data excluded

due to stray light contamination at the start of each of these months. The other cases with more than 5 missing days across all hours (e.g., May and December 2007 and 2008) are at least in part associated with extended instrument outages and in some

cases satellite outages, leading to the loss of multiple days of data. Apart from these cases, missing data are generally randomly distributed through the month and the specific days that are missing generally change from hour to hour. Hence, the effect of missing data on the monthly hourly averages may also affect the fidelity of the diurnal cycle in unexpected ways.

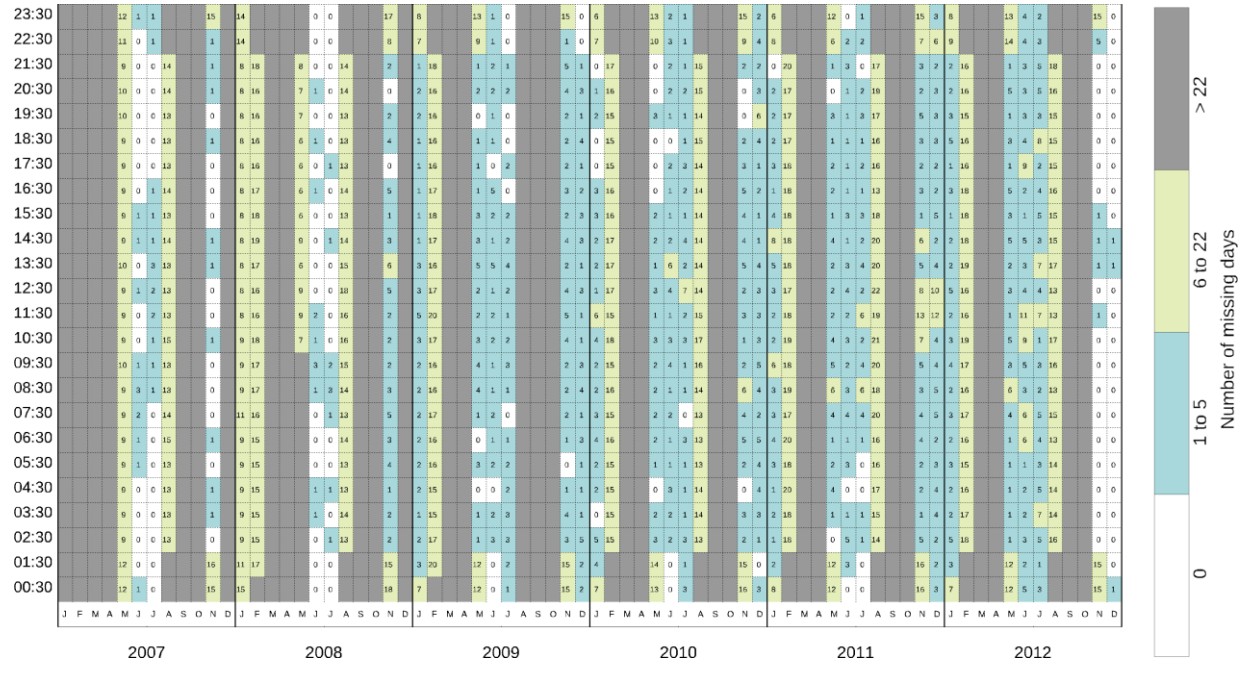

**Figure 2: Number of missing days of data per month as a function of hour and year. Cells are coloured according to number of missing days for that hour and month, with turquoise indicating 5 or fewer missing days, pale green between 6 and 22 days missing and grey more than 22 missing days. Where there are 22 or fewer missing days the actual number of days missing is indicated in the box. The colour divisions are chosen to highlight the hours with no missing days and delineate the data included in the unfilled and filled products as discussed in Sect. 3.**

## 2.3 Strategy for filling missing GERB data

Considering the amount of missing data in the GERB dataset and the effect this is likely to have on the monthly hourly average, it is clearly desirable to investigate methods to fill some of the missing information. Given the pattern of missing data, with multiple occurrences of several hours and indeed several days missing in some cases, filling the gaps by interpolating the existing GERB observations is not viable. Ideally an alternative source of information responsive to the meteorology present during the periods of missing data that can be used to fill in the gaps in the record is required.

The prime instrument on the Meteosat second generation satellites is SEVIRI. This instrument provides radiances in 11 narrow band channels from 0.635 to 13.4 μm every 15 minutes with a resolution of 3 km at the sub-satellite point. The GERB HR products, on which the Obs4MIPs dataset is based, are provided as a snapshot at the time of the corresponding SEVIRI observation, at a resolution of 3x3 SEVIRI pixels, on a grid aligned with the SEVIRI grid. As part of the GERB processing an empirical narrowband to broadband conversion is applied to the SEVIRI radiances to derive estimates of the broadband

radiances (Clerbaux et al., 2008a, 2008b). These so called 'GERB-like' radiances are converted to flux with the same conversion factor used to determine the GERB fluxes from the GERB radiances (Dewitte et al., 2008).

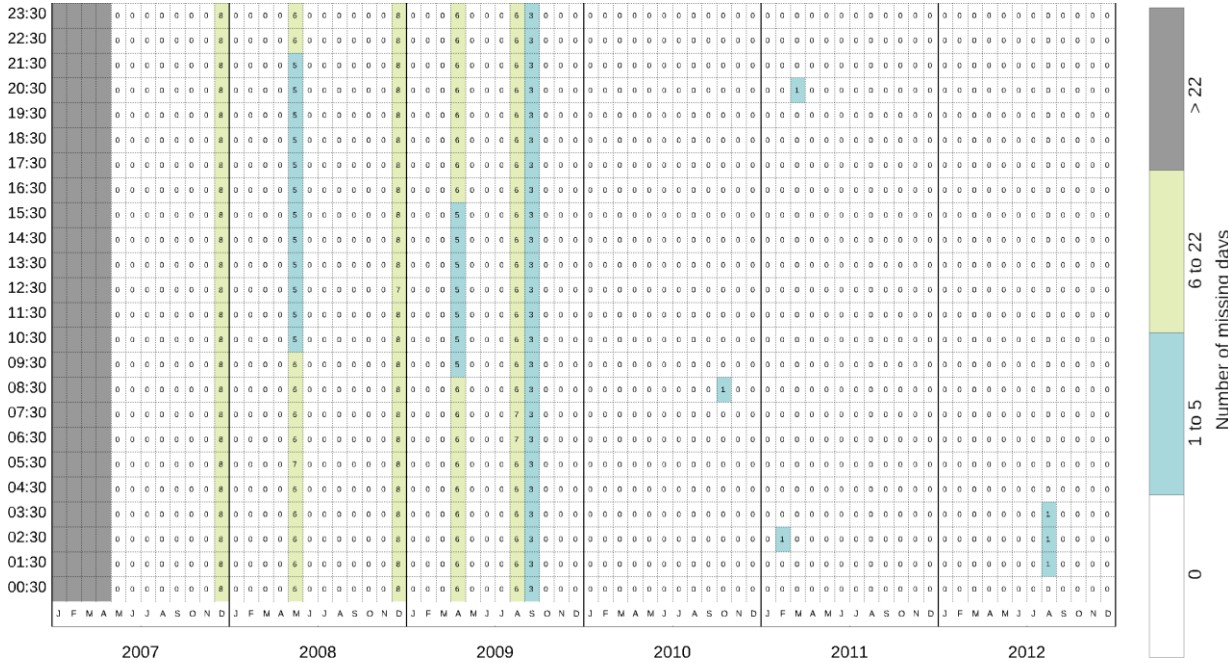

**Figure 3**: As **Figure 2** for GERB-like observations.

The SEVIRI based GERB-like fluxes suffer from significantly less missing data than the original GERB record (compare Fig. 3 and Fig. 2). Except for a few extended outages in the first few years which are a result of satellite level anomalies, nearly all the data missing in the GERB record are present in the GERB-like. Thus, the latter record may be useful for filling much of the missing GERB data.

The way the GERB-like fluxes are used in the GERB processing places no requirements on their absolute accuracy and limited requirements on their relative accuracy. Our expectation is that differences between GERB and GERB-like fluxes due to deficiencies in the narrowband to broadband conversion and due to the calibration of the original narrowband observations will need to be addressed before the GERB-like can be used to replace missing GERB data. Narrowband to broadband conversion errors will likely have scene and angular dependencies that do not vary a great deal over time, except in relation to
these variables. Conversely, calibration related errors would be expected, at first order, to manifest across different scenes in a similar, reproduceable way, but may vary in time. There may also be cross terms, where calibration changes manifest across the scenes differently due to variation in the weighting of the channels between scenes. For the GERB-like to be a suitable proxy for the GERB data, we need to understand and correct not just for the average offset between GERB and the GERB-like but must also account for the way the difference varies with scene, time of day and location.

Figure 4 and Fig. 5 show the spatially resolved monthly hourly mean GERB to GERB-like ratio for a selection of different UTC hours and months for the RSW and OLR respectively. The ratios shown in these figures are determined from the 1 degree

latitude/longitude monthly hourly averages constructed from the GERB and GERB-like fluxes, where the available data used to construct these averages has been matched in both data sets. GERB-like data are always present when the GERB fluxes are available as they are a required part of the GERB processing, so matching the data availability simply involves removing

GERB-like observations from the average where the corresponding GERB data are missing.

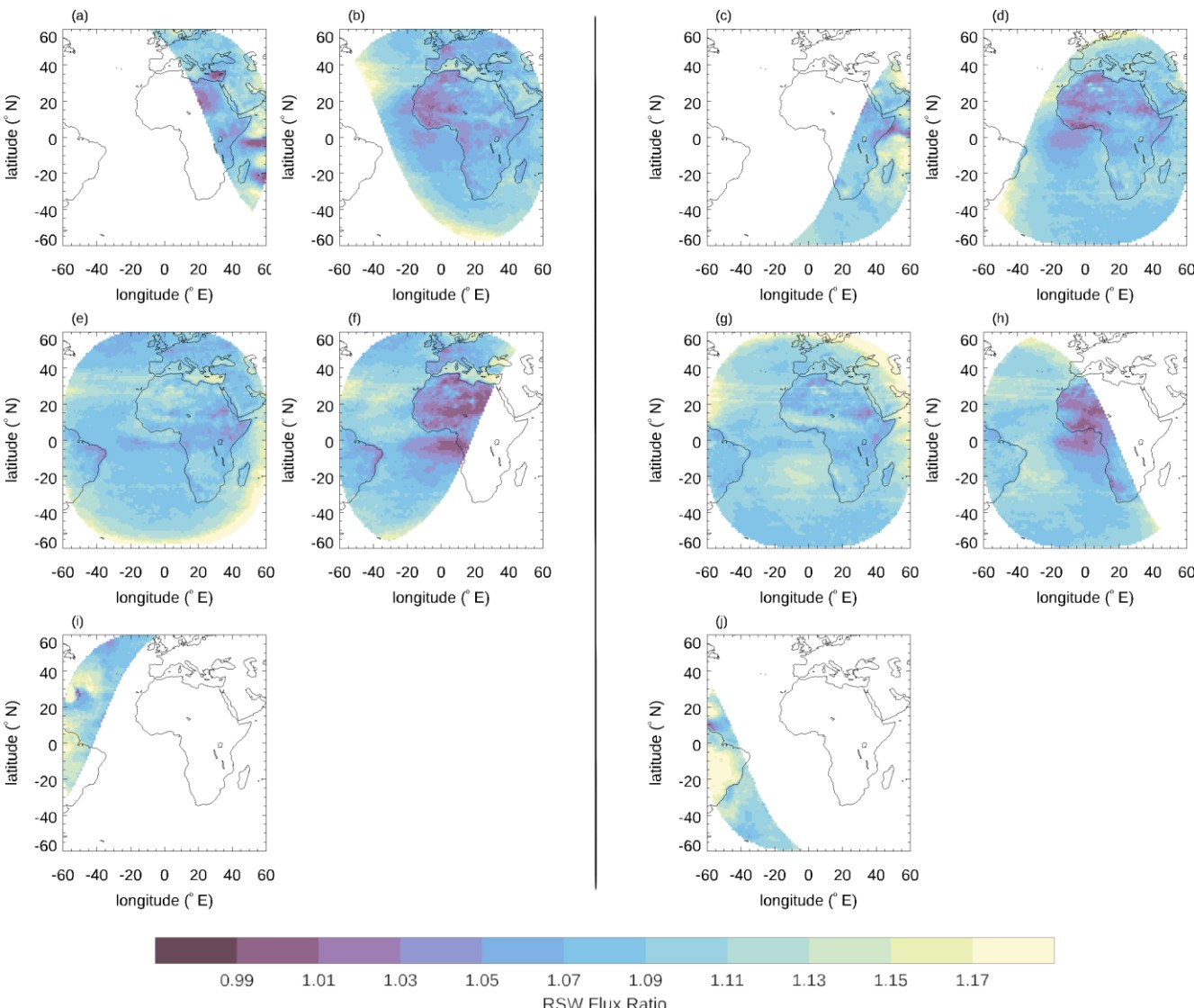

**Figure 4: GERB/GERB-like RSW ratio of the monthly hourly mean at 1 degree latitude/longitude scale, for June 2009 in the two left-hand columns (a, b, e, f, i) and December 2009 in the two right-hand columns (c, d, g, h, j) for 04:30 (a and c), 08:30 (b and d), 12:30 (e and g), 16:30 (f and h) and 20:30 (i and j) UTC.**

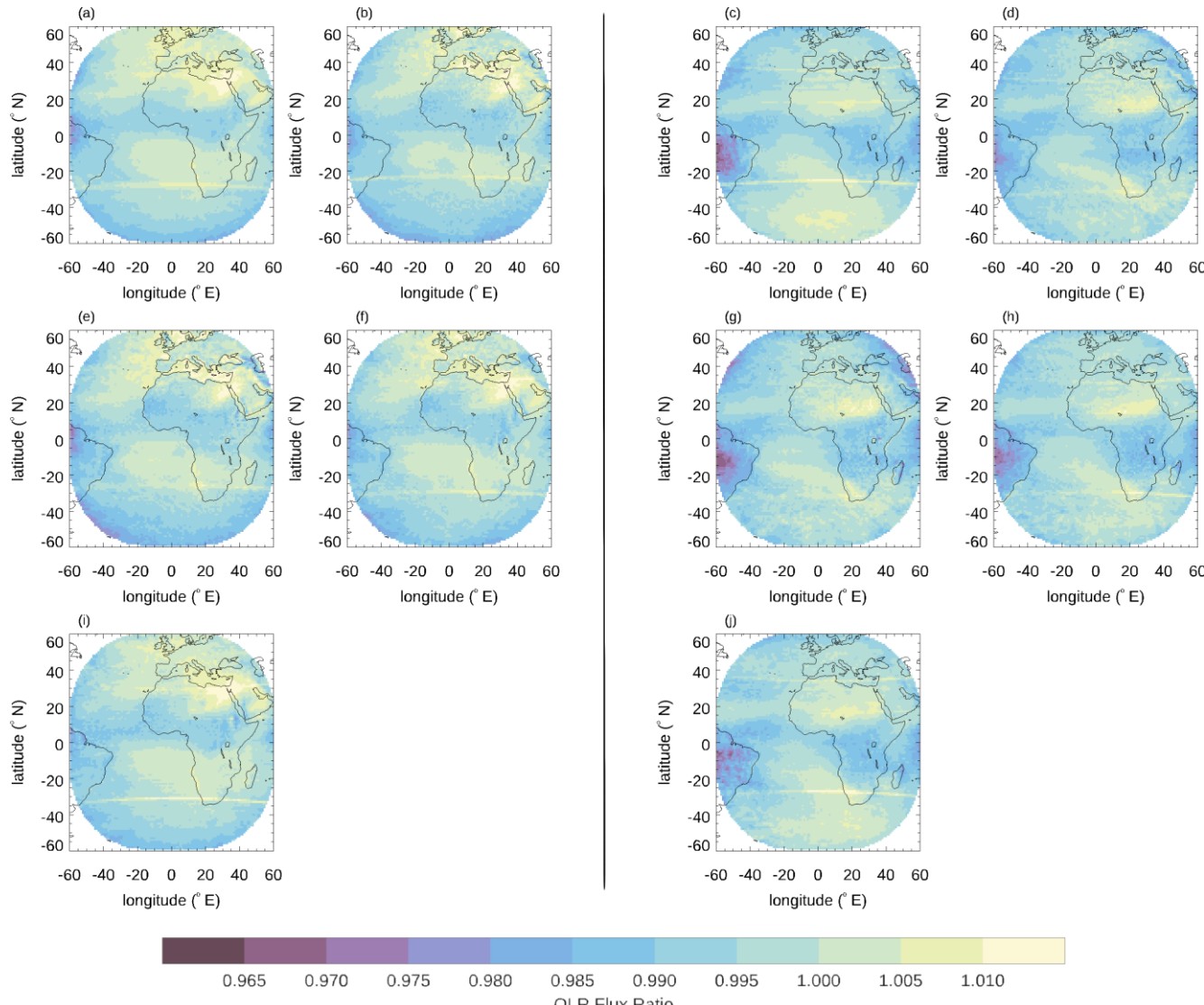


**Figure 5: As Fig. 4 but for the GERB/GERB-like OLR ratio for June 2009 in the two left-hand columns (a, b, e, f, i) and December 2009 in the two right-hand columns (c, d, g, h, j) for 04:30 (a and c), 08:30 (b and d), 12:30 (e and g), 16:30 (f and h) and 20:30 (i and j) UTC.**

The ratios shown in Fig. 4 and Fig. 5 illustrate both a global bias between the two sets of fluxes and angularly dependent

effects that manifest differently according to scene type. For the RSW, the ratio between the GERB and GERB-like fluxes

generally varies between 0.95 and 1.2. Variations occur with viewing and solar angle and thus with both location and time of

day and, more subtly, time of year associated with the variation of the solar zenith angle. The lowest RSW ratios tend to occur

at larger solar zenith angles over land. The highest RSW ratios occur over ocean and are mostly at larger solar zenith angles,

especially when combined with large viewing zenith angles. For the OLR the ratios are generally less extreme than the RSW,

with the lowest values of around 0.97 observed towards the edge of the GERB region at the largest viewing zenith angles for

the coldest scenes. The fixed viewing geometry of the geostationary platform means that viewing zenith angle effects correspond to fixed locations. The diurnal variation in the GERB to GERB-like OLR ratio is small and is associated with marked changes in scene, for example the daily heating of the land, seen most significantly over desert regions such as the Sahara. Similarly, seasonal variations in the OLR ratio are associated with scene variations such as the seasonal variation in
the positioning of the Intertropical Convergence Zone (ITCZ) and changes to solar induced land heating.

Figure 4 and Fig. 5 show that the ratio between the GERB and GERB-like fluxes does indeed exhibit a variety of the expected variations between the two datasets, with strong angular and scene dependent patterns in the ratio of the fluxes dominating. However, we find the day-to-day variation in the overall bias between the two datasets (not shown) manifests at a much lower level in both the OLR and RSW and is difficult to distinguish from the combined effect of scene dependent bias and day-to-
day variation in scene make up. If adjusting by the GERB/GERB-like ratio calculated at the monthly hourly mean 1 degree longitude/latitude scale can provide a good match between the GERB and GERB-like fluxes at the daily hourly scale, then the latter could be used to replace missing days of GERB data. Figure 6 displays the average and range of the mean and standard deviation of the individual daily hourly 1 degree longitude/latitude GERB – GERB-like difference distributions, as a function of UTC hour, before and after adjustment of the GERB-like. Results are shown for the RSW (left hand panels) and the OLR
(right hand panels) and summarize the individual distributions of the 1 degree longitude/latitude differences for each hour of every day where GERB and GERB-like data are available, as long as there are no more than 22 missing days in the month. As by definition, adjustment by the monthly ratio removes the monthly mean bias, the shift in the average value of the daily error distributions mean to around zero is expected. However, the reduction in the range of mean values after correction shows that the mean bias at the daily hourly level is consistently reduced by the monthly correction to less than a few W m$^{-2}$. Similarly,
the reduction in the standard deviations shows that despite day-to-day variations in meteorology, a correction derived at the monthly scale significantly reduces the range of errors seen at the daily hourly 1 degree longitude/latitude scale with the standard deviations reducing from an average of 10 to 4.6 W m$^{-2}$ in the RSW and from 2.2 to 1.7 W m$^{-2}$ in the OLR. These results demonstrate that a single monthly hourly correction applied at the 1 degree longitude/latitude scale significantly improves the fidelity between the GERB-like and GERB fluxes at the daily hourly scale.

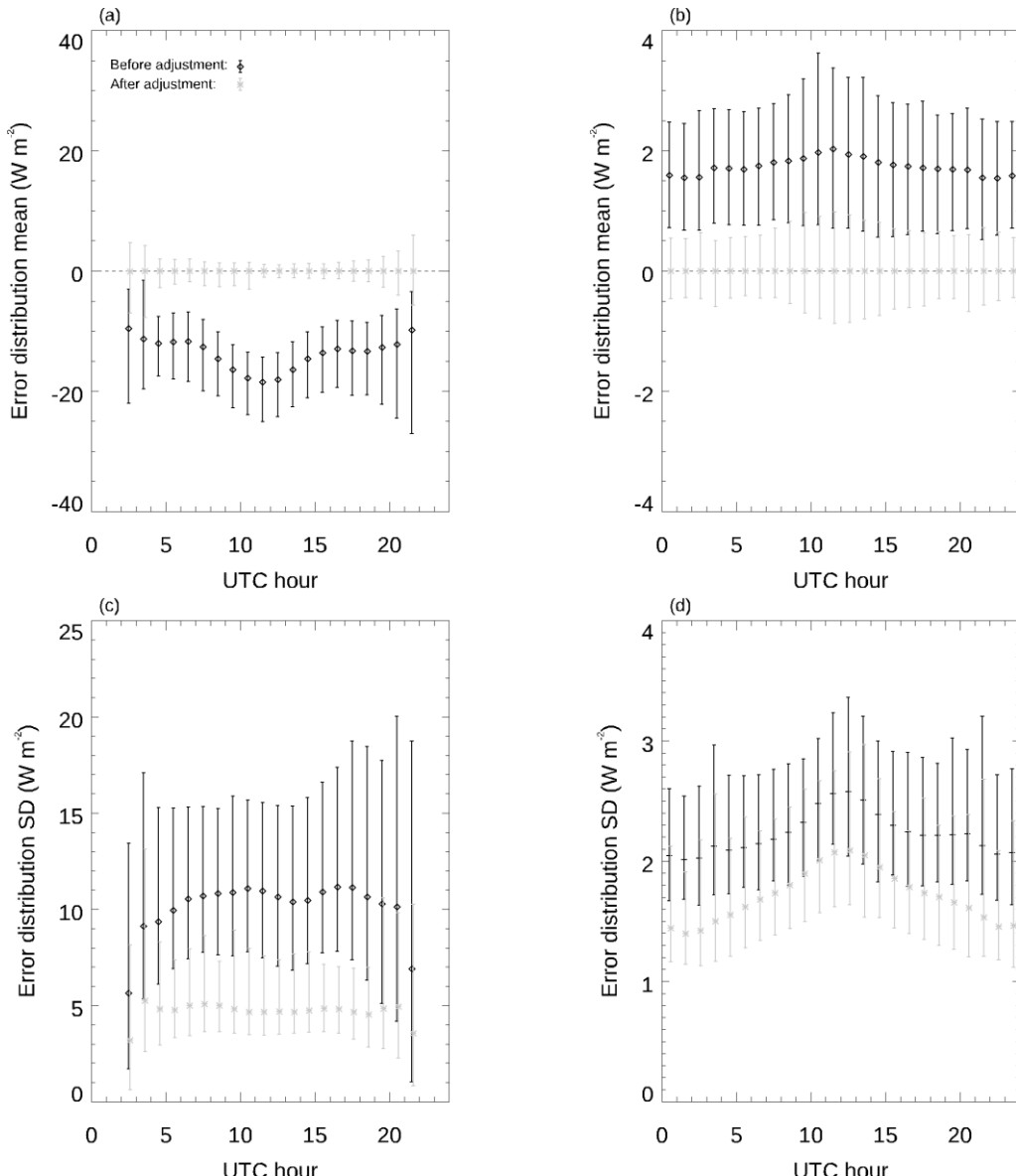

**Figure 6:** **Summary statistics for the GERB-like – GERB difference before (black) and after (grey) adjustment of the GERB-like by the monthly hourly ratio. Points indicate the average and the bars the range of these statistics over all the days at each hour. Results are shown as a function of UTC hour for the RSW (panels a and c) and the OLR (panels b and d) for the mean of the distribution (a and b) and the standard deviation (c and d). Times are on the half hour in all cases, but the plotting for the adjusted case is slightly offset on the x-axis for clarity.**

Thus, using corrected GERB-like data to fill missing hours of GERB data and then averaging over the month should improve the accuracy of the average. The required GERB-like correction is determined from the ratio between the GERB unfilled monthly hourly average and a corresponding GERB-like average calculated following the process outlined in Fig 1., with the GERB-like data used to determine the average matched to the GERB data availability. This provides a monthly correction at

the 1 degree longitude/latitude as a function of hour which is then applied to daily hourly GERB-like data. The corrected GERB-like daily hourly data are used to fill in missing hours of the GERB record before averaging over the month to produce filled GERB Obs4MIPs products. This process is illustrated in Fig. 7, in which the 1 degree latitude/longitude GERB and GERB-like hourly and monthly hourly products, referred to as 'hourly 1°x1°' and 'monthly-hourly 1°x1°', are derived following the steps outlined in Fig. 1.

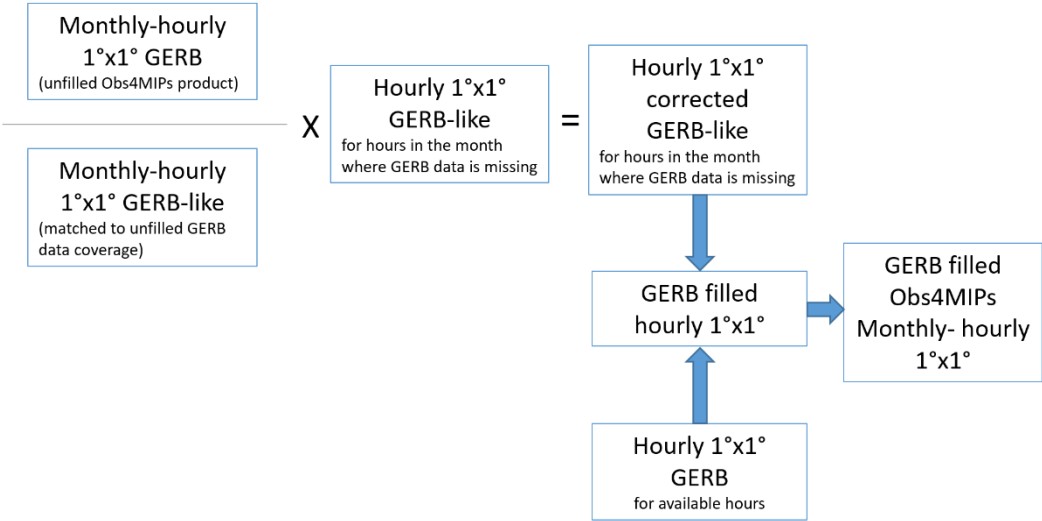


**Figure 7: Schematic illustrating how GERB-like data are corrected and used with the GERB hourly data to produce filled Obs4MIPs products. The monthly-hourly 1 degree latitude/longitude products denoted (1°x1°) here are produced using the steps illustrated in Fig. 1.**

## 3 Evaluation of the GERB Obs4MIPs monthly hourly average products

Whilst the instantaneous GERB data, including the HR products have been validated (Clerbaux et al. 2009; Parfit et al. 2016), the effect of missing GERB observations on the fidelity of the GERB filled and unfilled Obs4MIPs averaged products needs additional consideration. As illustrated in Fig. 2 there are a significant number of monthly hourly averages where one or more days of GERB observations are missing. These gaps at the daily hourly scale, if left unfilled, result in errors in the Obs4MIPs monthly hourly averages due to the uncaptured day-to-day variability in the fluxes. Alternatively, if these gaps are filled then

the impact on the monthly hourly average of the difference between the proxy data used for filling and the GERB data they represent needs to be assessed. In this section we provide estimates of these error sources as a function of number of missing days, considering the effect of both randomly distributed and consecutive missing days. In Sect. 3.1 we address how this impacts the V 1.0 unfilled GERB Obs4MIPs products originally released and in Sect. 3.2 we evaluate the error in the V 1.1 averages after filling.

## 3.1 Impact of missing data on the fidelity of the unfilled GERB Obs4MIPs products

For the unfilled GERB Obs4MIPs products the error in the monthly average due to missing data can be estimated by considering the effect of removing days from a month of data with complete, or nearly complete, coverage. Every UTC hour of the GERB-1 record with no more than one missing day during the month was used as a starting point for this analysis. This represents just over a third of the data for the months not affected by the systematic outages around the equinoxes. It also provides good coverage of the diurnal cycle for each of these months.

In this analysis, we consider each of the 'complete' or 'nearly complete' monthly hourly averages as the 'true' value. Differences between these true values and the averages calculated after the removal of a selected number of days provide an estimate of the error due to missing data. The effect of removing between 1 and 12 days randomly distributed through the month was calculated for eight different realisations of the days chosen. The effect of removing between 2 and 22 consecutive days was also determined for three different patterns: all days missing at the start of the month, at the end of the month and centred around the middle of the month.

Figure 8 displays example results for the removal of three randomly chosen days of data from the December 2012 11:30 UTC monthly hourly average. Four different realisations of the missing days are shown. The variation in the spatial distribution of the error (panels a to d for the RSW and e to h for the OLR) highlights the effect of the altered sampling. The largest differences in averages are seen for the RSW in the more strongly illuminated summer hemisphere and are for the most part associated with the averaging of synoptic variability at higher latitudes. Notable errors are also present in other regions which exhibit significant day-to-day variability in cloud coverage and/or properties, such as deep convective regimes over southern Africa. For both the OLR and RSW the detail of the spatially resolved errors varies for each of the realisations depending on the meteorology on the individual days removed. However, the overall distribution of errors shown in panel i for the RSW and panel j for the OLR is relatively stable from realisation to realisation. For both the OLR and RSW the distributions are relatively symmetrical about the mean, which is close to zero. As might be anticipated from the spatial error patterns, the spread in the error is significantly larger for the RSW than the OLR with the associated standard deviations between 3.5 to 4 times higher for the former. We will use the mean and standard deviation of the error distribution as summary statistics for interpreting the change in the errors as a function of number of days missing, time of day and month.

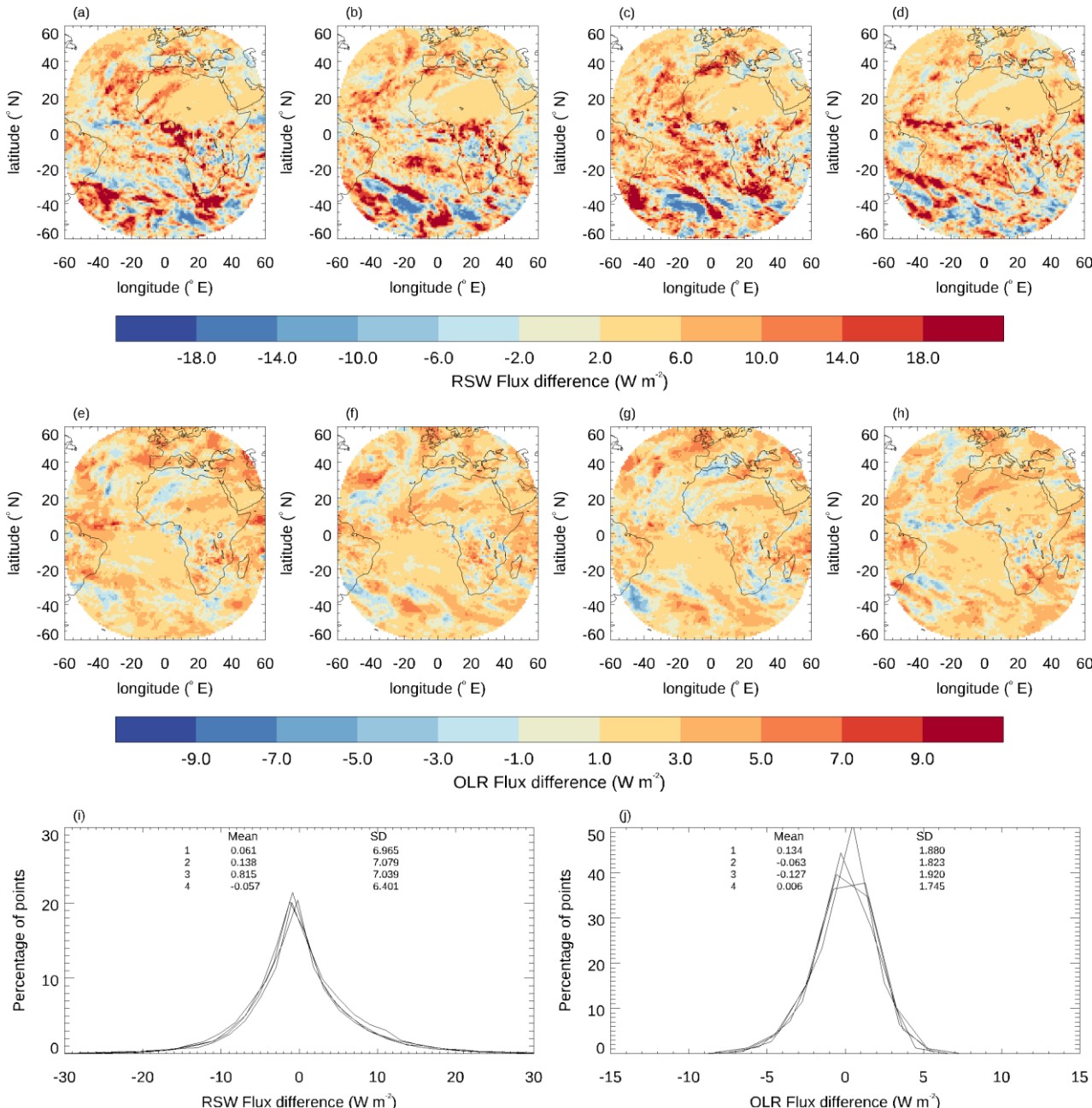

**Figure 8: Impact on the 1 degree latitude/longitude monthly hourly mean fluxes of removing 3 randomly chosen days of data from December 2012 11:30 UTC. The results for four different random realisations of the days removed are shown spatially resolved for RSW in panels a, b, c and d and for OLR in panels e, f, g and h. The corresponding distributions of the flux difference are shown for RSW and OLR in the bottom panels i and j respectively with the mean and standard deviation in each of the four cases also displayed.**

Considering the results for all months and times of days used in this analysis we find that for the OLR, systematic variations in the standard deviation and mean of the resulting error distribution, both seasonally and diurnally, are small and difficult to

distinguish from the variability resulting from the choice of days. Seasonal variation in the error distribution is also negligible for the RSW, aside from a small reduction in variability in the standard deviation and a very slight reduction in its value for July. This is associated with an increasingly dominant contribution from the Sahara which has low day-to-day variability. However, there is a notable diurnal signal in the standard deviation of the RSW error distribution. Even when only calculated over the locations which are not in twilight or night-time at any point in the month at that hour, the standard deviation, which is relatively stable between 10:30 and 15:30 UTC when there is a high level of solar illumination, drops steadily for earlier and later times of day, due to the overall reduction in the incoming solar flux. Results for hours earlier than 04:30 and later than 19:30 UTC are more unpredictable and generally noisy as there are typically less than 20 % of the full number of points represented in the statistics due to the limited portion of the disc illuminated at these times. Thus, for the RSW, combining results for all months and for the hours 10:30 to 15:30 UTC, gives an indication of errors at the height of the disk illumination. Errors at 04:30 and 19:30 UTC represent the error distribution for the low illumination case, when there are still a sufficient number of points illuminated to obtain reasonable statistics.

Figure 9 summarizes the expected monthly hourly mean error due to missing data at the 1 degree scale, in terms of the standard deviation and mean of the error distribution for both randomly and systematically removed days. The results show that on average the mean and standard deviation increase roughly linearly as the number of missing days increases. The variability in the standard deviation and mean also increases as the number of missing days increases but in a less regular manner. For the 10:30 to 15:30 UTC time range the standard deviation of the RSW error distribution increases rapidly as the number of missing days increases, exceeding 10 W m$^{-2}$ for some cases with four or more consecutive missing days or five or more missing days randomly distributed through the month. The corresponding standard deviation which is exceeded for the OLR in these cases is 3 W m$^{-2}$. For the mean of the error distribution, which is the overall image bias due to the missing data, individual realisations can see increasingly large biases as the number of missing days increases. When consecutive days are removed the bias may exceed 2 Wm$^{-2}$ for the RSW and 1 W m$^{-2}$ for the OLR for as few as 3 or 4 missing days for some of the cases.

To avoid averages with unacceptably large errors, monthly hourly averages are only provided for the V 1.0 unfilled GERB Obs4MIPs release when there are 5 or fewer missing days data in the month for that hour. This means that the V 1.0 GERB Obs4MIPs monthly hourly data is limited to the hours and months shaded in white or turquoise in Fig. 2 (a total of 645 monthly hourly averages), with the hours of each month shaded yellow or grey not provided to users for these products.

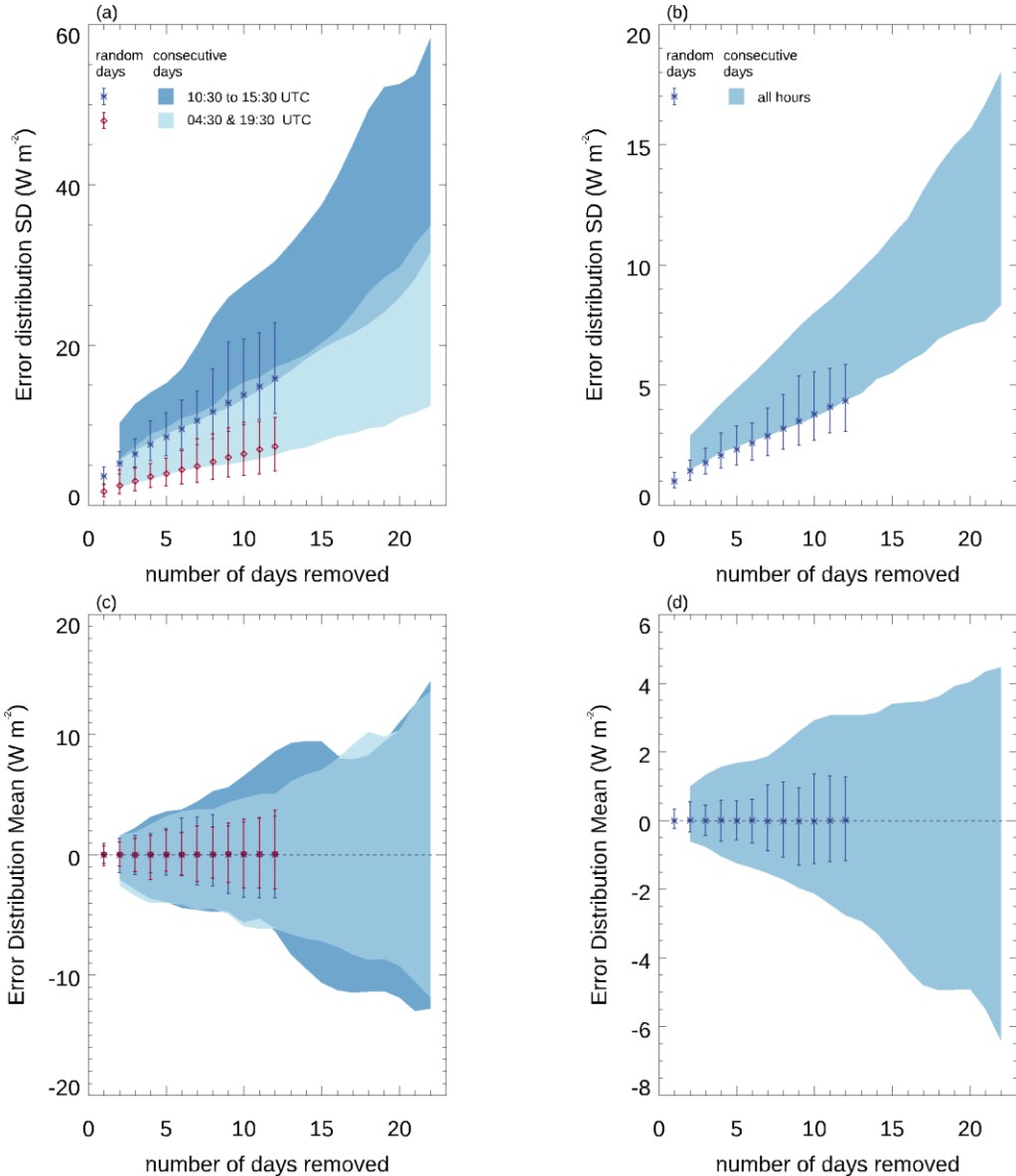

**Figure 9: Summary statistics for the error distribution in the monthly hourly mean 1 degree latitude/longitude fluxes due to missing days. Standard deviation (panels a and b) and mean (panels c and d) of the error distributions are shown as a function of number of days removed for the RSW (a and c) and OLR (b and d) fluxes. The average and range over the realisations and months are shown for days removed chosen at random as points with bars. The corresponding range for points systematically removed from various points in the month is shown as shaded regions. For the RSW results are shown separately for the UTC hours 10:30-15:30,**
**representing high solar illumination of the GERB region, and for 04:30 and 19:30 combined, representing low solar illumination. The OLR results are shown for all times together.**

### 3.2 Fidelity of the filled GERB Obs4MIPs products

Whilst the improvement in correspondence between the GERB and GERB-like daily hourly fluxes after adjustment with the

monthly hourly ratio discussed in Sect. 2.3 is encouraging, these results are not quite representative of the situation in the case

of missing GERB data. In this case the monthly hourly ratio derived from incomplete GERB and corresponding GERB-like fluxes will need to be used to correct GERB-like fluxes that are not included in that average. Thus, for the adjusted GERB-like to be useful for filling missing GERB data, it needs to be shown that rescaling by a monthly hourly average ratio derived from incomplete data can sufficiently improve the GERB-like fluxes at the daily-hourly 1º scale for the missing periods. Analogous to the approach used in Sect. 3.1, starting with all the hours of the record with no more than one missing day of

GERB data in the month, we determine the effect of removing increasing amounts of GERB data and replacing it with GERB-like data scaled by the monthly hourly ratio. In each case we match the data-coverage for both GERB and GERB-like: i.e., corresponding points are removed from both data records before calculating the monthly hourly means and the associated ratio. As for the unfilled average comparison described in Sect. 3.1, the error due to filling can then be estimated from the difference between the resulting filled average and the average calculated from the GERB data alone before any data were removed.

Figure 10 summarises statistics of the residual error at the monthly hourly average 1 degree latitude/longitude scale for the filled data. It can be directly compared to Fig. 9 which shows the equivalent results for the unfilled averages. Comparing the two figures shows that filling the missing days of GERB fluxes with their scaled GERB-like equivalents before calculating the monthly hourly average, reduces both the mean and standard deviation of the error in the monthly hourly average at the 1 degree scale by more than a factor of 10 in all cases. Given these improved statistics we implement this filling approach to

produce our 'filled' GERB Obs4MIPs product and use it in the next section to perform an initial evaluation of climate model performance. We note that the level of error reduction is retained even when there are up to 22 days systematically missing, and thus we are also able to reinstate the months of February and August in the filled record. Therefore, filled GERB monthly hourly Obs4MIPs products can be provided to users for all hours of the month that are not shaded grey in Fig. 2 with the error associated with filling bounded by the values shown in Fig. 10. This results in 1030 monthly hourly averages available to users

of the V 1.1 filled GERB Obs4MIPs products compared to the 645 for the V 1.0 unfilled products.

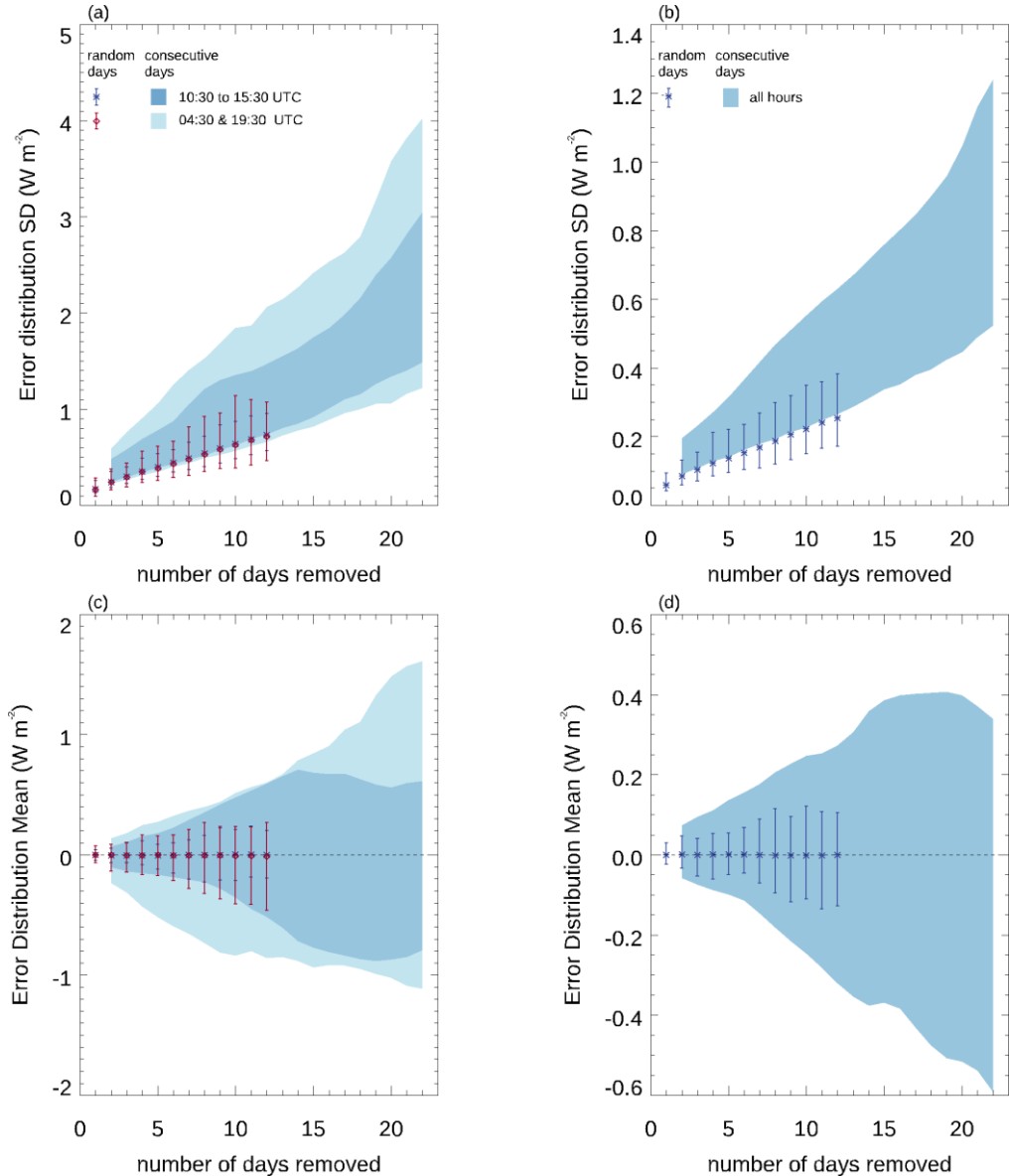

**Figure 10: As Fig. 9 but for the error distribution in the monthly hourly mean 1 degree latitude/longitude fluxes due to filling missing days with scaled GERB-like data as described in the main text. Note the change in y-axis scales compared to Fig. 9.**

## 4 Application of the GERB Obs4MIPs filled product to climate model evaluation

Top of the atmosphere radiative fluxes are routinely used as an evaluation metric for climate model performance, with model parameters often tuned to produce a realistic radiation budget. This is typically performed at a relatively coarse temporal and spatial scale (monthly or global annual means) which has the potential to mask compensating errors. A more stringent test, at least at the process level, is to compare temporally resolved fluxes. This type of comparison has also been recognised as

potentially insightful for assessing cloud feedback (Webb et al., 2015) and has led to a limited number of modelling centres starting to produce and archive monthly hourly mean top of atmosphere radiative fluxes from Atmospheric Model Intercomparison Project (*amip*, Gates 1992) type runs. Here we compare such fluxes, as simulated by two versions of the climate configuration of the Hadley Centre Global Environmental model HadGEM3, with the V 1.1 filled GERB Obs4MIPs product. Concentrating on two cloud regimes we show how the diurnally resolved fluxes can complement other observationally based evaluations and provide unique insights into the model fidelity.

## 4.1 HadGEM3 configurations and simulation description

Our analysis concerns historical *amip* simulations of two different Global Coupled configurations of HadGEM3, (GC3.1 and GC5.0). Both model configurations consist of atmosphere, land, ocean and sea ice subcomponents, have 85 vertical layers and are run at N96 (1.875º longitude by 1.25º latitude) horizontal resolution. The *amip* simulations are forced with observations of sea-surface temperatures, sea-ice cover and historical forcings (Eyring et al., 2016).

GC3.1 is the configuration that underpinned the United Kingdom's contribution to CMIP6 (Williams et al., 2018; Mulcahy et al., 2018; Walters et al., 2019). The most recent configuration (GC5.0) has not been documented yet but includes three changes affecting cloud that are particularly relevant to our analysis. A prognostic-based convective entrainment linked to surface precipitation which introduces memory into the convection scheme is expected to improve the representation of the diurnal cycle of convection over land. A new bimodal diagnostic cloud fraction scheme (Van Weverberg et al., 2021a & 2021b), and a reformulation of the 'cloud erosion' term (Morcrette, 2012) in the large scale cloud scheme (Wilson et al., 2008a & 2008b), are expected to improve the realism of cloud evolution and increase the amount and optical thickness of low-level cloud, particularly in the subtropics and lower mid-latitudes.

Monthly mean diurnal cycles of TOA radiative fluxes (all-sky and clear-sky) are produced for the entire length of the *amip* experiment. The radiative fluxes are hourly means, centred, as in the observations, on the half-hour, and the monthly mean diurnal cycle is constructed by averaging each UTC hourly mean over the entire month. These diagnostics were requested for the *amip* experiment of phase 3 of the Cloud Feedback Intercomparison Project (Webb et al., 2017). The HadGEM3 OLR diagnostics used in this study differ from those submitted to CFMIP3. The OLR diagnostics submitted to CFMIP3 contain a correction that accounts for the surface temperature adjustment by the boundary layer scheme in model time steps between radiation time steps. This OLR diagnostic adjustment is introduced to conserve energy, but it significantly distorts the diurnal cycle of OLR (its impact on daily and longer time averages is very small). Given that this OLR correction is purely diagnostic (i.e. it doesn't affect the model evolution) and it was not designed to work on sub-daily timescales, here we have used the OLR without this correction.

## 4.2 Model evaluation

For the purposes of highlighting the utility of the V 1.1 GERB Obs4MIPs product we focus on two cloud regimes, marine stratocumulus and deep convection. Improving the representation of sub-tropical stratocumulus has been a focus for climate

modellers for some time due to its importance in determining global cloud feedback (e.g. Bony and Dufresne, 2005). In general, models have tended to simulate too little marine stratocumulus, with what is present being too bright (e.g. Nam et al., 2012). In the multi-annual mean, Williams and Bodas-Salcedo (2017) report good agreement between GC3.1 and CALIPSO height-frequency statistics over stratocumulus, but with a distribution that shows too few moderately optically thick clouds,

compensated by too many optically thick clouds. Comparisons with CERES-EBAF monthly mean top-of-atmosphere RSW fluxes imply that this translates into stratocumulus decks that are too reflective.

Deep convective regions continue to present a challenge at least in part because of the scale at which convection is typically parameterised in global climate models (e.g. Guichard et al., 2004, Hohenegger and Stevens, 2013, Christopoulos and Schneider, 2021). Although improvements have been made (e.g. Stratton and Stirling, 2012), a persistent issue over land is

that convective clouds tend to rain out too early, leading to too little cloud in the late afternoon to evening, when deep convection (and precipitation) typically peaks in observations (e.g. Yang and Slingo, 2001, Tan et al., 2019). Such issues persist to some extent even in higher resolution simulations (e.g. Watters et al. 2021). Given the temporal resolution of the GERB Obs4MIPs product it is ideally suited to investigate whether adjustments to the parameterisations that affect convective invigoration and lifecycle in GC5.0 are having a beneficial impact in terms of the top-of-atmosphere energy budget.

We begin with a qualitative comparison of the overall monthly means to provide context to the diurnally resolved regional comparisons that follow. Figure 11 shows decadal average monthly mean January RSW fluxes as simulated by GC3.1 (a) and GC5.0 (b) over the region 60°S – 60°N and 60°E – 60°W. V 1.1 GERB Obs4MIPs RSW fluxes are shown in panel (c), in this case averaged over the five years of GERB-1 January observations. The corresponding information for June is shown in panels (d)-(f), with, in this case six years of observations available for averaging. Broadly speaking the simulations capture the patterns

seen in the observations, including the seasonal shift in the positioning and strength of features such as the ITCZ and stratocumulus decks off Angola and Namibia. There are differences: during the summer hemisphere GERB shows significantly higher RSW fluxes over the highest latitudes. It is noticeable that GC5.0 also tends to be brighter than GC3.1 in those regions. GC5.0 also appears to show more extensive, brighter marine stratocumulus off the west African coast in both seasons compared to GC3.1.

Equivalent information to Fig. 11 is shown in Fig. 12 for OLR fluxes. In this case the most obvious differences between the two HadGEM3 simulations are located in regions of tropical deep convection. In June GC5.0 appears to shift the peak of convection within the ITCZ further east. In January the centres of deep convection over Brazil and central southern Africa are both strengthened in GC5.0 relative to GC3.1. Visually, both changes appear more in line with the GERB observations, although the intensity of land convection still appears greater in the observations.

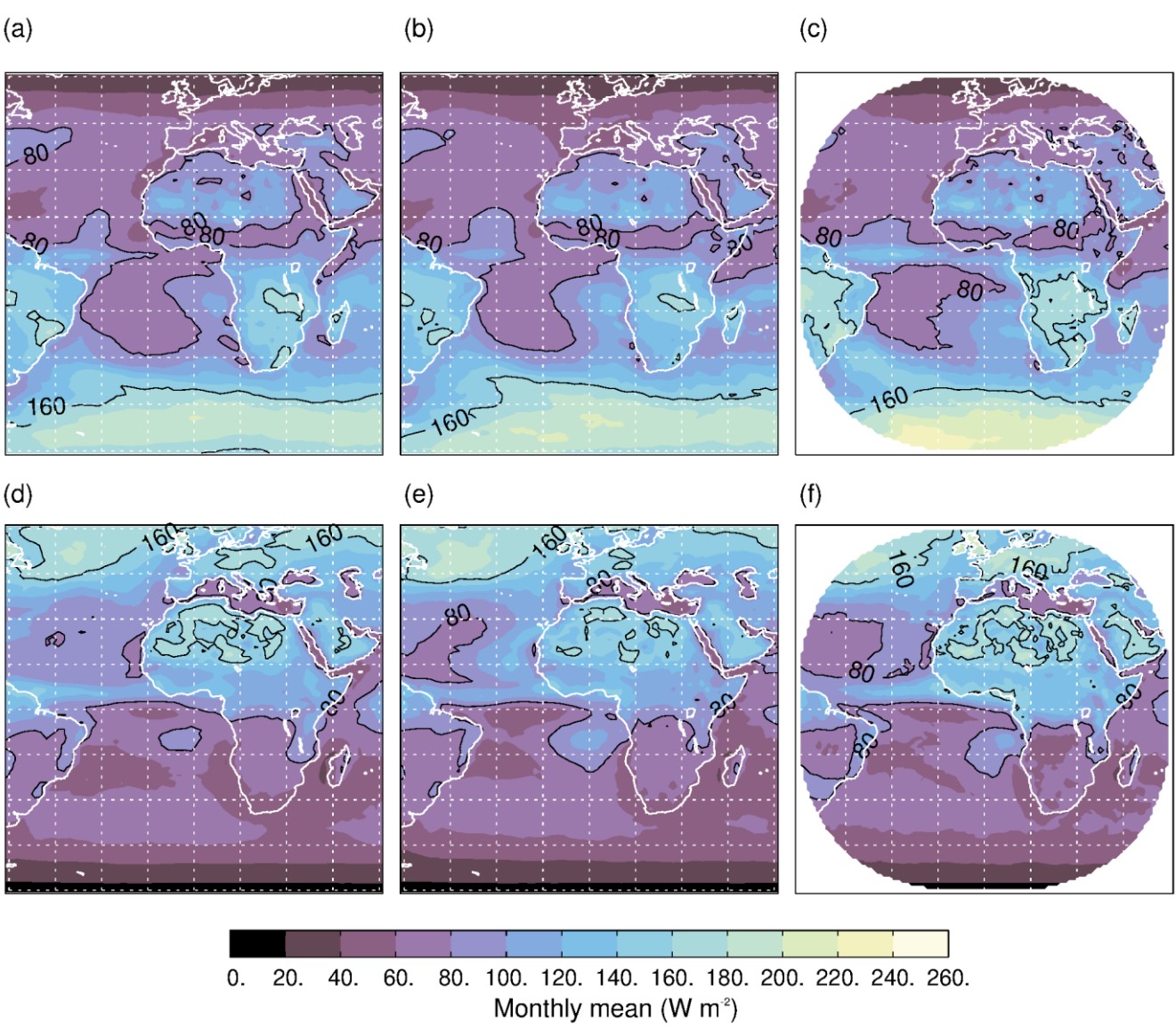


**Figure 11: Monthly average top of atmosphere RSW fluxes for January in the top row (a, b and c) and June in the bottom row (d, e, and f) from GC3.1 (left panels a and d), GC5 (middle panels b and e) and V 1.1 GERB Obs4MIPs (right panels c and f). Simulated fluxes are a decadal mean (2000-2009). GERB fluxes are averaged over the duration of the GERB-1 observations.**

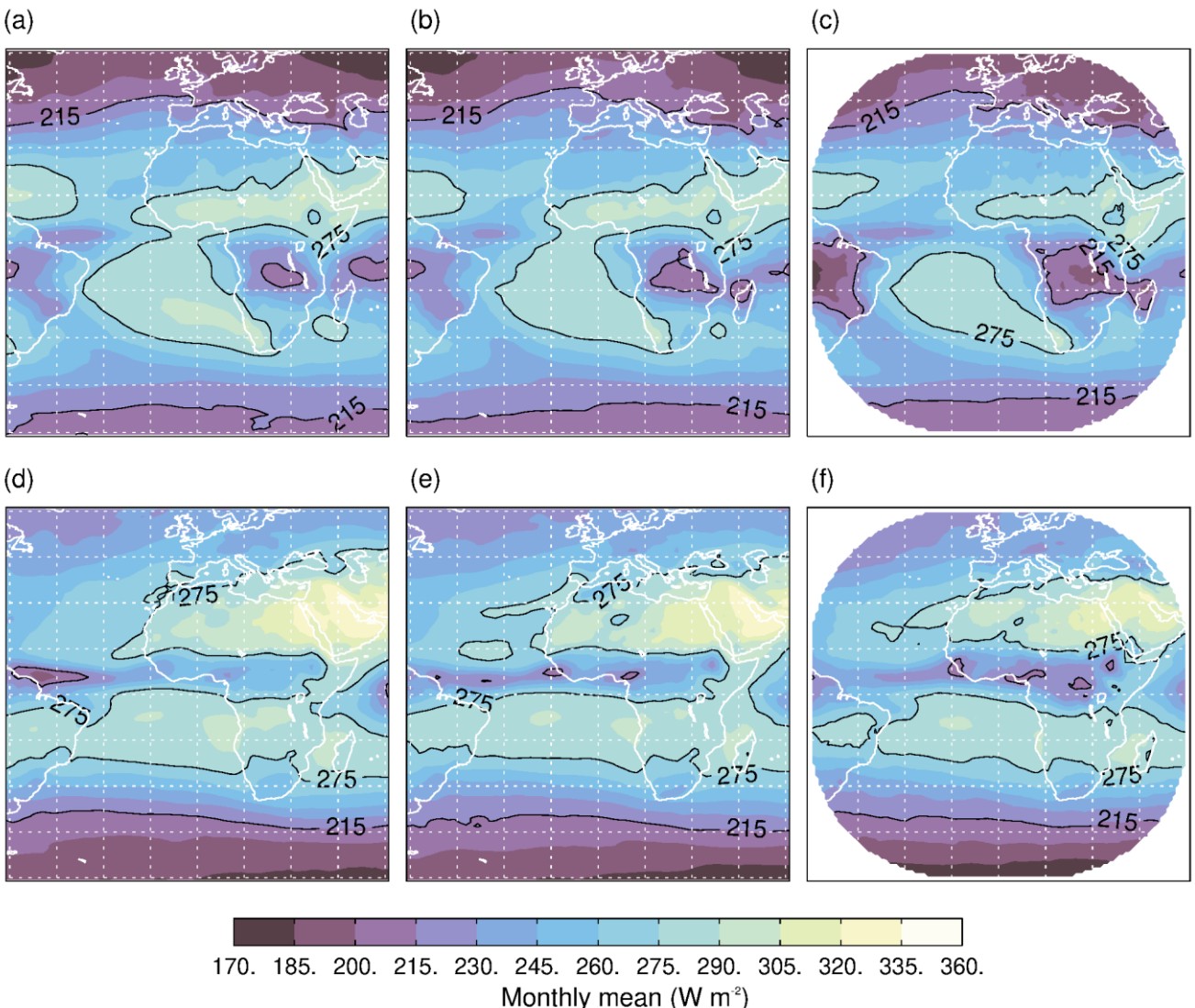

 **Figure 12: As Fig. 11 for OLR fluxes.**

To provide a more quantitative analysis we define two seasonally dependent latitude-longitude boxes encompassing the south-east Atlantic stratocumulus deck and African deep convection. Table 1 shows the multi-year June and January monthly mean fluxes obtained from both sets of simulations and from GERB in these regions. We note that shortening the period of averaging in the simulated datasets to be commensurate with the length of the GERB record makes a difference of, at most, 3 W m$^{-2}$ in

the mean fluxes.

Over the stratocumulus region Table 1 reinforces the qualitative impression from Fig. 11 and Fig. 12, with the change in HadGEM3 configuration resulting in a distinct brightening in both June and January. In June, the degree of brightening means that the mean RSW flux exceeds that measured by GERB, whereas in January, the increment is still insufficient to reach the

level of the observed fluxes. As might be anticipated given typical stratocumulus altitudes, the impact on the OLR fluxes is
less marked but consistent between the months, reducing by of the order 3 W m$^{-2}$. In concert, these two results imply an enhanced cloud fraction, optical depth or both in the GC5.0 configuration.

| | South Atlantic Marine Stratocumulus | | | | African Deep Convection | | | |
|---|---|---|---|---|---|---|---|---|
| | June (-16-10° E, 3-22°S) | | January (-16-10° E, 3-28° S) | | June (14-37° E, -2-12° N) | | January (15-31° E,0-17° S) | |
| | RSW | OLR | RSW | OLR | RSW | OLR | RSW | OLR |
| GERB | 76.8 | 283.9 | 94.4 | 275.1 | 129.5 | 228.6 | 161.6 | 208.3 |
| GC3.1 | 67.6 | 287.4 | 82.1 | 284.2 | 105.3 | 260.2 | 139.7 | 227.3 |
| GC5.0 | 82.1 | 284.8 | 92.5 | 281.2 | 106.5 | 253.4 | 141.6 | 221.6 |

**Table 1: Multi-year June and January monthly mean RSW and OLR fluxes over regions characterised by marine stratocumulus and deep convective cloud as observed by GERB and simulated by the two configurations of HadGEM3 outlined in the main text.**

The largest differences between the two sets of simulated fluxes over deep convection are realised in the OLR. Moving from
GC3.1 to GC5.0 results in a reduction in OLR of order 7 W m$^{-2}$ in both months, while a small increase of less than 2 W m$^{-2}$ is seen in the corresponding RSW fluxes (Table 1). These changes move the GC5.0 fluxes towards the observations but there is still a notable overestimate in OLR and corresponding underestimate in RSW flux, particularly in June, consistent with the visual impression of 'missing' land convection in the simulations during this month (Fig. 12).

To understand the reasons behind the changes in the model fluxes in both regions we use diagnostics produced by version 1.4
of the CFMIP (Cloud Feedback Model Intercomparison Project) Observational Simulator Package (COSP; Bodas-Salcedo et al., 2011). In particular, we use vertical profiles of cloud fraction of the Cloud–Aerosol Lidar and Infrared Pathfinder Satellite Observation simulator (CALIPSO), and International Satellite Cloud Climatology Project (ISCCP) histograms of cloud fraction in intervals of cloud-top pressure (CTP) and cloud optical thickness ($\tau$). The CALIPSO and ISCCP simulators are documented in Chepfer et al. (2008) and Klein and Jakob (1999), respectively.

Figure 13 illustrates these diagnostics for January. Results for June are qualitatively similar. GC5.0 shows a significant increase in cloud fraction in the stratocumulus region (Fig. 13 a), with clouds also being optically thicker (Figs. 13 c and e). These two changes contribute to the increase in RSW described above. In the deep convection region, GC5.0 shows an enhanced cloud fraction at high altitudes, coupled with lower cloud top height (Fig. 13 b). The impact of these two changes on the OLR will partially cancel out. However, GC5.0 also shows optically thicker clouds (Figs. 13 d and f). The combined increase in cloud
fraction and optical thickness leads to a reduction in OLR in GC5.0 compared to GC3.1 (Table 1), despite the reduction in cloud top height.

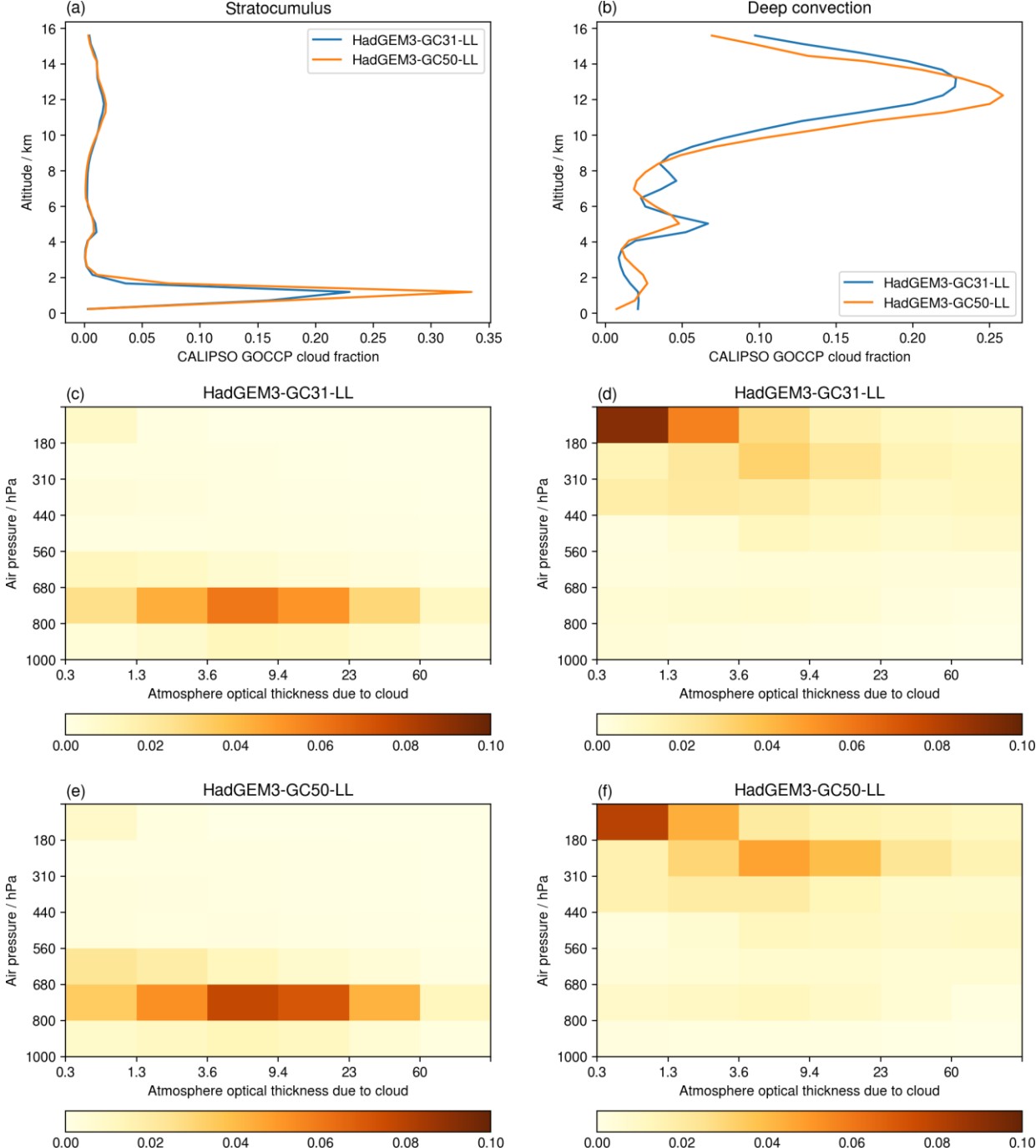

**Figure 13. Multi-annual monthly average cloud fraction for January. Vertical profiles of COSP/CALIPSO cloud fraction (a and b), and COSP/ISCCP CTP-τ histograms of cloud fraction (c to f). The left columns show plots for the stratocumulus region and the right column for the deep convection region.**

Utilizing the diurnally resolved V 1.1 GERB Obs4MIPs fluxes we analyse these results further by decomposing them as a function of time of day. Figure 14 shows the regional hourly monthly mean RSW fluxes from each HadGEM3 configuration for each individual year of the simulation as well as the 10 year model mean over the stratocumulus regions. Superposed in colour are the GERB Obs4MIPs fluxes for 2007-2012. Figure 15 shows equivalent information for OLR fluxes over the regions
of deep convection.

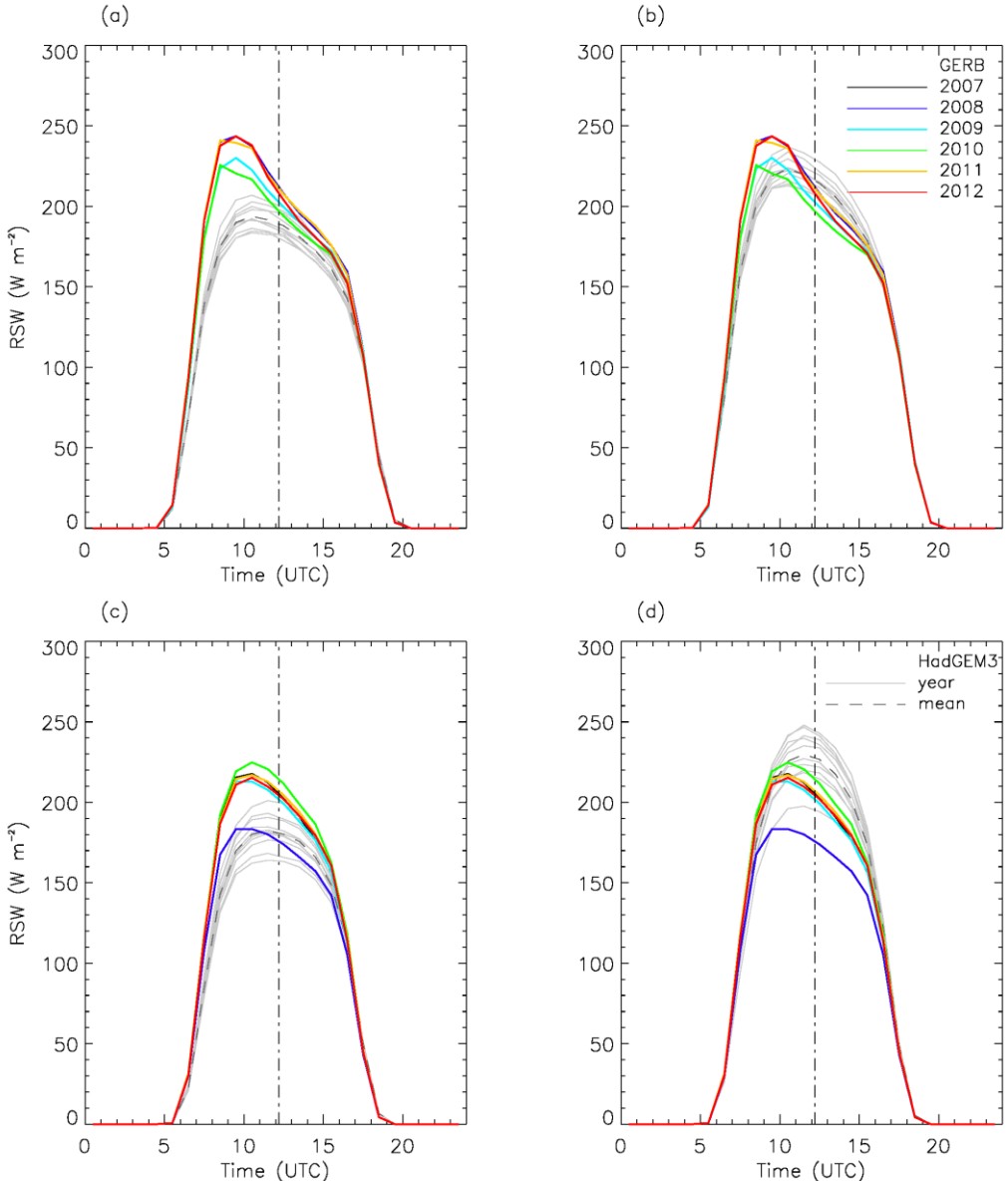

**Figure 14: Monthly hourly mean RSW fluxes over the marine stratocumulus regions identified in Table 1 for January (a, b) and June (c, d). Coloured lines show GERB Obs4MIPS fluxes for each year of the GERB observations. Solid grey lines show simulated fluxes for each simulation year and the dashed grey line the 10 year mean for the HadGEM3-GC3.1 (a, c) and HadGEM3-GC5.0 (b,**
**d) configurations. Dot-dashed vertical lines show the approximate timing of local noon.**

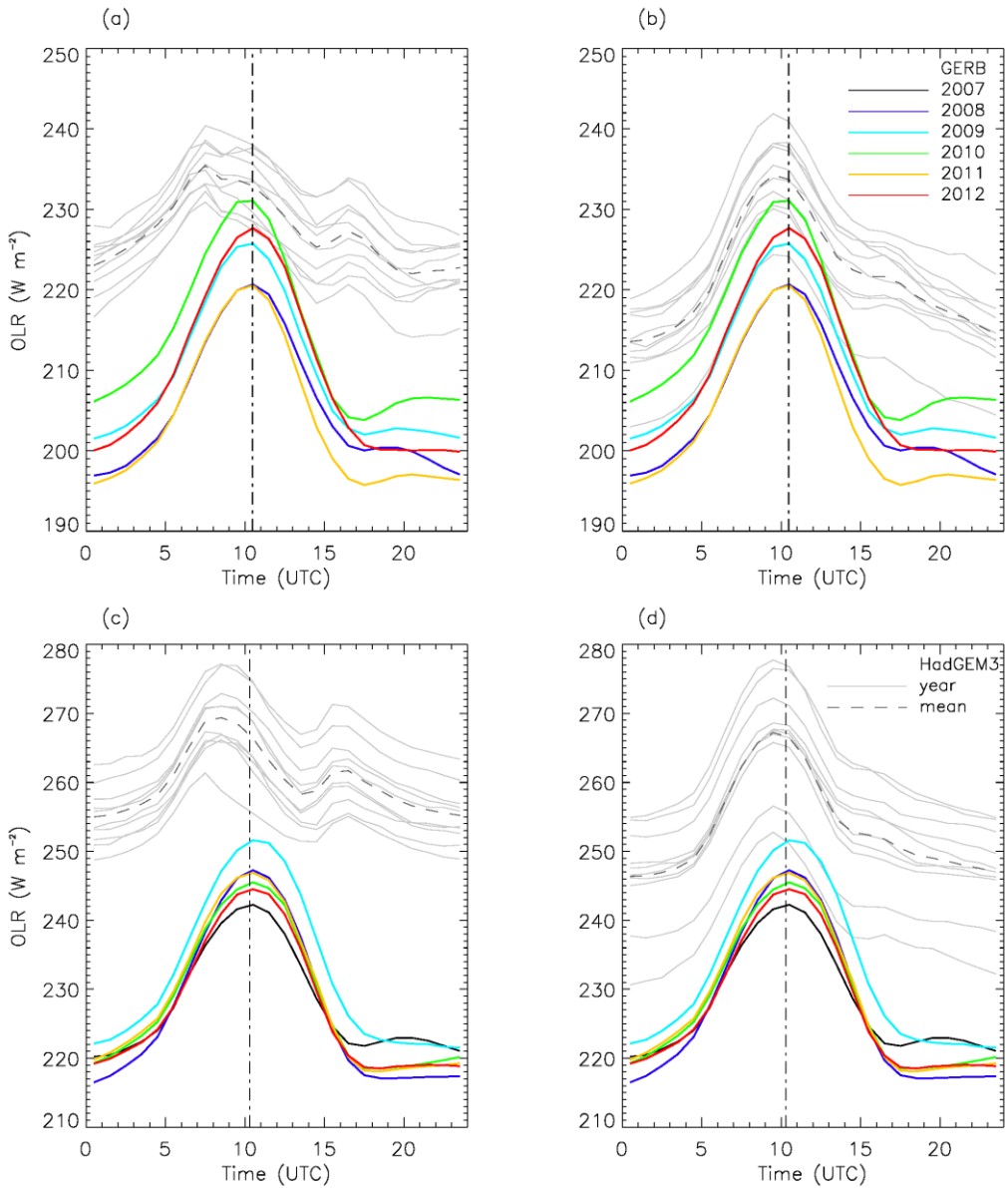

**Figure 15: As Fig. 14 for monthly hourly mean OLR fluxes over the deep convective regions identified in Table 1 for January in the top row (a and b) and June in the bottom row (c and d). Simulations for GC3.1 are shown in panels (a) and (c) and for GC5.0 in panels (b) and (d).**

Focusing on Fig. 14 first, the observations show the classic signature of stratocumulus development and thickening in the morning prior to decay through the afternoon, manifested as a clear asymmetry in the RSW fluxes around local noon (e.g. Gristey et al., 2019). This asymmetry is more pronounced in January than June. There is significant year-to-year variability in

the magnitude of the observed fluxes (peak values can vary by up to 40 W m$^{-2}$) but they all have this characteristic phasing. The degree of observed inter-annual variability is smaller in January than June – behaviour that is also captured in the model simulations. While the simulations do exhibit a diurnal asymmetry they are unable to fully capture its observed magnitude. Similarly, although they show a constant diurnal phase from year-to-year, peak values for both model configurations are typically delayed by an hour compared to the observations (Table 2). However, comparison of the GC3.1 and GC5.0

configurations does reinforce the impression that within these limitations, the latter is able to better capture the observed behaviour even if the improvement to the phasing between the configurations is slight.

Turning to the deep convective regions (Fig. 15), the observed OLR fluxes show a spread over the years considered which reaches of the order 10-15 W m$^{-2}$. The phasing of the cycle changes between the two months, with OLR fluxes reaching their maximum just after local noon in June and just before or at local noon in January. For both months the timing of the maximum

is consistent from year to year, although there is marked interannual variation in the shape of the cycle towards late afternoon and eve3ning, particularly in January. The corresponding simulated values from GC5.0 highlight an improved ability to capture the general shape of the diurnal cycle, with the removal of what appears to be a spurious secondary peak in the OLR fluxes in late afternoon in GC3.1. The timing of the OLR maximum is shifted later in GC5.0 by between 1-2 hours and is more consistent with the observations, albeit still too early in the day. The amplitude of the cycle is also improved (Table 2). These

improvements in the diurnal cycle are mainly driven by the introduction of the prognostic entrainment rate. Clearly other issues remain: in June the fluxes are consistently too high implying either missing convection or convection which is not vigorous enough. The interannual variability over the region is significantly higher than seen in the observations, which would be consistent with this interpretation. Both issues are present to a lesser extent in January. However, overall, the direction of travel from GC3.1 to GC5.0 is encouraging, particularly when viewed in a diurnally resolved comparison.

In climate models, the diurnal cycle of convection is typically evaluated using the diurnal cycle of precipitation (e.g. Stratton and Stirling, 2012). The remote sensing technology, spatio-temporal sampling and retrieval algorithms used in the precipitation retrievals introduce substantial uncertainty in the timing of the maximum of precipitation in the mean diurnal cycle (Dai et al., 2007; Minobe et al., 2020). The GERB dataset presented here provides a very accurate description of the monthly mean diurnal cycle of the OLR and RSW fluxes, making it an excellent tool for the evaluation of the diurnal cycle of convection in models.

It is worth noting that the minimum in OLR is delayed by around 3h with respect to the maximum in precipitation in convective regions (Dai et al., 2007), and therefore a combination of radiation and precipitation diagnostics can provide a more detailed picture of the evolution of precipitation and the anvil cloud associated with the development of deep convection.

| | South Atlantic Marine Stratocumulus (RSW) | | | | African Deep Convection (OLR) | | | |
| --- | --- | --- | --- | --- | --- | --- | --- | --- |
| | June (-16-10°E, 3-22°S) | | January (-16-10°E, 3-28°S) | | June (14-37°E, -2-12°N) | | January (15-31°E,0-17°S) | |
| | Amp | Phase | Amp | Phase | Amp | Phase | Amp | Phase |
| GERB | 135.0 | 10:30 | 141.0 | 09:30 | 17.7 | 10:30 | 16.9 | 10:30 |
| GC3.1 | 114.8 | 11:30 | 111.7 | 10:30 | 9.2 | 08:30 | 8.1 | 07:30 |
| GC5.0 | 147.6 | 11:30 | 130.8 | 10:30 | 13.8 | 09:30 | 12.6 | 09:30 |

**Table 2: Amplitude and phase in multi-year June and January monthly mean RSW and OLR fluxes over marine stratocumulus and deep convective regions, as observed by GERB and simulated by the two configurations of HadGEM3. Amplitude, A, is defined as $A = Max(x_t - \bar{x}_t)$ where $x_t$ is the RSW or OLR flux as a function of hour through the day, and phase is the time (in UTC) at which the value of A is realised.**

## 5. Data availability

The V 1.0 unfilled and V 1.1 filled GERB Obs4MIPs OLR and RSW products presented in this paper are available from the Centre for Environmental Data Analysis (https://doi.org/10.5285/7aa17e66aaab4ece87064272b9f94e3a (Bantges et al. 2021a) and https://doi.org/10.5285/4fa633d24d104217a4c9d3fb3589f35d (Bantges et al. 2021b) for the V 1.0 (unfilled) OLR and RSW and https://doi.org/10.5285/90148d9b1f1c40f1ac40152957e25467 (Bantges et al. 2023a) and https://doi.org/10.5285/57821b58804945deaf4cdde278563ec2 (Bantges et al, 2023b) for the V 1.1 (filled) OLR and RSW). The datasets are also available on the Earth System Grid Federation.

The characteristics of the GERB Obs4MIPs products are summarized in Table 3.

| Variable | Monthly hourly TOA OLR flux | Monthly hourly TOA RSW flux |
| --- | --- | --- |
| Dataset name | GERB-HR-ED01-1-0.1hrCM.rlut (unfilled) GERB-HR-ED01-1-1.1hrCM.rlut (filled) | GERB-HR-ED01-1-0.1hrCM.rsut (unfilled) GERB-HR-ED01-1-1.1hrCM.slut (filled) |
| Variable name | rlut | rsut |
| Absolute accuracy of underlying data | 1% | 2.25% |
| 1-sigma uncertainty in the monthly hourly average at the 1 degree latitude/longitude scale due to missing/filled data | < 3.2 W m$^{-2}$ (V 1.0) < 1.3 W m$^{-2}$ (V 1.1) | < 12 W m$^{-2}$ (V 1.0) < 3 W m$^{-2}$ (V 1.1) |
| Variable type | 1hrCM monthly means of hourly mean data | |
| Spatial resolution | 1 degree latitude/longitude grid centred on the half degree | |

| | |
|---|---|
| **Temporal resolution** | Monthly hourly, monthly means of hourly mean data centred on the half hour UTC |
| **Valid region** | The region approximately 60°N to 60°S, 60°E to 60°W (presented on a global grid) |
| **Available months** <br> Months in italics are only available in filled V1.1 release | 2007: *May*, June, July, *August*, November <br> 2008: *January*, *February*, *May*, June, July, *August*, November <br> 2009: January, *February*, May, June, July, November, December <br> 2010: January, *February*, May, June, July, *August*, November, December <br> 2011: January, *February*, May, June, July, *August*, November, December <br> 2012: January, *February*, May, June, July, *August*, November, December |
| **Data format** | Climate and forecast (CF) version 1.7 compliant netCDF |

Table 3: Data characteristics of the GERB Obs4MIPs products.

Model outputs used for the comparisons presented in Sect. 3 are available at https://doi.org/10.5281/zenodo.10101394.

## 6. Conclusions

The GERB Obs4MIPs products are specifically designed to enable the evaluation of the diurnal cycle in top of the atmosphere
radiation fluxes, as simulated by climate and Earth-system models. This paper has described in detail how the GERB Obs4MIPs products are derived from the baseline GERB HR data to give monthly hourly mean OLR and RSW fluxes on a regular 1 degree latitude/longitude grid. Whilst the instantaneous GERB data have been fully evaluated and compared against the CERES products in previous comparisons (Clerbaux et al. 2009; Parfit et al. 2016, Doelling et al. 2013, 2016), because of the relative prevalence of missing observations, which occur both randomly throughout the record and systematically around
the equinoxes, particular attention has been paid in this study to the impact of missing data on the fidelity of the averages. Our results show how estimates of the instantaneous broadband 'GERB-like' fluxes from the SEVIRI narrowband instrument can be used to fill missing GERB data. A scaling factor is calculated from the ratio of the monthly hourly 1 degree latitude/longitude  averages for the available GERB and matched GERB-like data and applied to the daily hourly GERB-like data. Using these scaled GERB-like fluxes to fill the missing GERB observations at the daily hourly scale before averaging
significantly improves the fidelity of the monthly hourly averages when there are missing days of GERB data. For a given number of missing days the residual uncertainty in the monthly hourly average at the 1 degree latitude/longitude scale due to filling is more than a factor of 10 smaller than the error in the unfilled data due to missing days. Even when there is a substantial amount of systematic missing data, as the case for GERB in the months of February and August every year, using the scaled GERB-like data to fill the missing periods leads to relatively small errors which are comparable to the error manifested in the
unfilled dataset if just one day of data is missing. Using this method V 1.1 filled GERB Obs4MIPs products have been produced which provide greater coverage of the year and higher fidelity averages than the original V 1.0 unfilled products.

We use the new V 1.1 filled GERB Obs4MIPs products to perform a preliminary evaluation of two sets of *amip* type simulations for the HadGEM3 climate model. The two sets of simulations differ in their atmospheric components, with the newer configuration implementing a prognostic based entrainment rate scheme; a bimodal cloud scheme within entrainment zones associated with strong temperature inversions, and improvements to the influence of dry air entrainment on cloudy grid boxes. At the monthly mean level, there are noticeable differences in top-of-atmosphere fluxes, with an overall brightening in the newer GC5.0 configuration and an apparent strengthening of convection. Although such changes would be evident in comparisons with existing radiative flux observations, further decomposing into the monthly hourly diurnal cycle allows insight into the amplitude and phasing of, in particular, different cloud regimes. Focusing on stratocumulus decks off south-western Africa and deep convection over Africa, the GERB Obs4MIPs product indicates that the monthly mean changes are consistent with an improved diurnal amplitude and, in the case of the convective region, phase in these regions. Discrepancies still remain: for example, the simulated RSW asymmetry seen over the stratocumulus deck is not as pronounced as in the observations and tends to be delayed by around 1 hour compared to the observations, for both model configurations. Similarly, deep convection over Africa in boreal summer is too weak, and in both the winter and summer seasons it tends to occur slightly too early resulting in an earlier simulated peak in OLR than seen in the observations. Tying these initial results to the behaviour of the underlying driving fields will be one avenue for future investigation.

We have shown that the GERB Obs4MIPs product is a very valuable complement to the traditional climatological averages of TOA radiation used for model evaluation. It provides a more direct connection with the model processes that control errors at both weather and climate timescales. Also, the fact that it is presented in CF-compliant netCDF format makes it extremely user-friendly, and ready to be incorporated into standard model evaluation tools like ESMValTool (Eyring et al., 2020).

Unfilled (V 1.0) and filled (V 1.1) GERB Obs4MIPs monthly hourly averages have been released as v 1.7 CF compliant netCDF products for the GERB 1 (Meteosat-9) observation period (May 2007 to December 2012). These are presented at 1 degree latitude/longitude resolution on a global grid with valid fluxes for the geographical region approximately 60°N – 60°S, 60°E – 60°W. Users are recommended to use the V 1.1 release for all applications. The V 1.1 products are available for 8 months of the year (January, February, May, June, July, August, November, December) for most of the released period. The underlying absolute accuracy of the GERB data is 1% for the OLR and 2.25% for the RSW, additional errors due to filling missing data are estimated to be less than 1.3 Wm$^{-2}$ for the OLR and less than 3 Wm$^{-2}$ for the RSW in V1 .1 monthly hourly averages at the 1 degree latitude/longitude scale. Obs4MIPs monthly hourly average products for the GERB 2 (Meteosat-8) period (May 2004 to February 2007) are currently in production using the V 1.1 methods described here and expected to be released soon. The short record and data quality issues affecting the GERB 3 (Meteosat-10) record (May 2015 to February 2018) as a result of various operational issues make it difficult to determine at this time if these data will be suitable for similar treatment. However, Obs4MIPs products for the GERB 4 (Meteosat-11) period (May 2018 to February 2023) are expected to be produced once the underlying data have completed the full record calibration stability assessment that is currently underway.

## Author contributions

The original draft manuscript was prepared by JER, HEB and RJB with substantial contributions to Sect. 3 from ABS. JER was responsible for developing the methodology of the GERB monthly hourly average product production and its filling. JER and RJB performed the error analysis related to missing data and data filling. RJB produced the software to generate the GERB Obs4MIPs dataset and produced the datasets needed to perform the error analysis. ABS provided the HadGEM3 model output and COSP analysis and contributed expertise on the interpretation of the model-data differences. HEB carried out the comparisons between the HadGEM3 and GERB Obs4MIPs data. Updates in response to reviews were led by JER with contributions from other authors where required and have been reviewed by all authors.

## Competing interests

None of the authors have any competing interests.

## Acknowledgements

The authors would like to acknowledge the efforts of the whole GERB team, with specific thanks to Edward Baudrez and Nicolas Clerbaux for facilitating the use of the GERB-like products. Special mention should also be made for the efforts of Joanna Futyan and Richard Allan as members of the GERB science team for their contribution to studies on the methods for filling of sun glint and twilight conditions for the released GERB products, without which it would not have been feasible to determine monthly averages from the GERB data.

Acknowledgement is also made to the funding support of the National Centre for Earth Observation (HEB and RJB) that enabled the production of the GERB Obs4MIPs datasets and to the Met Office Hadley Centre Climate Programme funded by DSIT (ABS).

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
