# Peer review of "The GERB Obs4MIPs: a dataset for evaluating diurnal and monthly variation in top of atmosphere radiative fluxes in climate models"

_Earth System Science Data, 2024_

## Author Comment (AC1)

**Reply to reviewer 1**

Our thanks to the reviewer for their helpful suggestions. In response to their overarching comment :

*"Overall, my opinion is that a more focused description of the data, highlighting strengths and weaknesses and presenting quantitative assessment of the uncertainty would improve the impact of this dataset. Although interesting, I am not sure the illustration of the dataset's use in evaluating a climate model simulation is central to a dataset description paper and would benefit from a more detailed exploration elsewhere. However, if the authors prefer this more lengthy discussion, I do not see any scientific reason to object and I consider that this work is suitable for publication with only minor modifications listed below."*

We think that the changes made to what was section 2, into a data production (new section 2) and data evaluation (new section 3) sections, more clearly emphasises the assessment of the dataset. We prefer to keep the inclusion of the brief comparison between model and data to highlight the utility of the dataset and demonstrate the information it can provide to study diurnally varying processes, such as the examples used.  We agree that further papers could certainly explore this further but would not wish to remove the discussion included in this paper.

We provide answers to the specific questions below.

*1) Title - I would not describe GERB Obs4MIP as a tool? How about something like:*

*"The GERB: a dataset for evaluating diurnal and monthly variation in top of atmosphere radiative fluxes in climate models."*

Answer: Title has been changed as suggested.

*2) L8 - since GERB data has been used in model evaluation for 20 years it is not new. What is new is the Obs4MIP aspect ("newly reprocessed" may be more appropriate).*

Answer: We have changed 'new' to 'newly available'. We choose not to use 'newly reprocessed' to avoid any implication that the underlying processing of the GERB data has changed. The GERB Obs4MIPs data presented are newly derived higher level products based on the long existing processing of the  GERB data.

*3) L11 "approximately" since it is not a square 60S-60N, 60W-60W domain*

Answer: Added as suggested.

*4) L19 ":" --> "," (or we could compromise with a ";")!*

Answer: Altered to a comma as suggested

*5) L20 improved fidelity, relative to GC3.1*

Answer: Amended as suggested

*6) L27 Since outgoing longwave is used later it may be worth stating here e.g. "emitted thermal infrared (outgoing longwave) and solar reflected..."*

Answer: Amended as suggested

*7) L36 also Allan et al. (2011), Examination of long-wave radiative bias in general circulation models over North Africa during May–July. Q.J.R. Meteorol. Soc., 137: 1179-1192. https://doi.org/10.1002/qj.717*

Answer: Added as suggested.

*8) L55 add a line pointing to the fact that the HR product is a resolution enhanced version of the GERB broadband radiative fluxes using SEVIRI narrow band measurements.*

Answer: As suggested the following sentence has been added:

> The GERB HR fluxes are a temporally interpolated, resolution enhanced version of original GERB observations, derived using spatial information on the scene variation within the GERB footprint from the SEVIRI imager.

*9) L63 "are" --> "were"*

Answer: Altered as suggested

*10) L72 why is 70 degrees zenith the cut-off and not slightly more or less?*

Answer: The limit follows the recommendation of the GERB quality summary beyond which errors increase and mapping to other grids becomes problematic. The sentence has been amended to indicate that the vza limit relates to the recommended valid range of the GERB products when averaging the data.

*11) L74 "proceed without prejudice" is not really clear what is meant (sounds like legal speak). Is one observation representative of a 100x100km area? Presumably there are mostly lots of values per 1x1 degree box.*

Answer: When there is no missing data a 1x1 degree box will, depending on its location, contain between 6 and 169 HR points for each of the four 15-minute time steps that comprise an hour. However, there are occasions when one or more of the 15-minute slots is missing, there are also much rarer occurrences when some of the HR pixels within the region are missing. In both these cases averages are made over the available points as long as there is at least one within the 1x1 degree box and hourly bin. Whilst in theory this means that one HR point could represent a whole hour of the day for the entire 1x1 degree region this is highly unlikely and if it occurs at all would likely be restricted to cases close to the edge of the valid region where there are fewer HR points to start with. We have added a sentence to explain this.

*12) L76 incoming solar flux from what dataset or algorithm?*

Answer: The process described reduces to an adjustment of the solar zenith angle. In practice we use the incoming solar flux used in the production of the GERB-like products. This assumes a solar constant of 1366 modulated by the calculated Earth-Sun distance for that day of the year and multiplied by the cosine of the solar zenith angle according to location. However, the assumed values of incoming solar and earth sun distance are irrelevant to the resulting product, as they do not change between the conversion from flux to albedo and back from averaged albedo to flux. Thus, the process is equivalent to adjustment to the solar zenith of the midpoint of the grid point and hour, with the overall level of flux maintaining that associated with the GERB data, and thus traceable to the observations themselves. The following sentences have been added to make this point clear:

> As the total solar irradiance and the Earth-Sun distance do not change during the conversion to albedo, and back to flux, this becomes purely an adjustment in solar zenith angle to the centre of the grid box and hour bin. The process is equivalent to multiplying each flux by the ratio $\cos(\theta_{local}) / \cos(\theta_{centre})$, where $\theta_{local}$ is solar zenith angle at the HR pixel time and position and $\theta_{centre}$ is the solar zenith angle at the 1 degree latitude/longitude centre at half past the hour.

*13) Figure 3 - the colour bar seems to bear no resemblance to the maps. Also this figure could be designed in a way to maximise the size of the panels (or reduce the size of the plot and all the dead space)*

Answer: Our apologies an incorrect scale was included. This has been amended and the panels have been rearranged to make better use of the available space as suggested.

*14) L187 "exiting" --> "existing"*

Answer: Amended as suggested

*15) L200 this paragraph could be reduced to the first line plus "(compare Figures 5 and 3)".*

Answer: We agree the paragraph should be reduced but suggest as a compromise it be reduced to the following 3 sentences (note figure numbers have changed due to rearranging of these sections):

> The SEVIRI based GERB-like fluxes suffer from significantly less missing data than the original GERB record (compare Fig. 3 and Fig. 2). Except for some extended outages in the first few years which are a result of satellite level anomalies, nearly all the data missing in the GERB record are present in the GERB-like. Thus, the latter record may be useful for filling much of the missing GERB data.

*16) L205 these lines are difficult to understand. Can they be written more clearly? There is a lot of detail in this section that may be unnecessary for users of the dataset so another option is to more briefly note issues and refer to prior papers if the reader is interested.*

Answer: We agree and have replaced this paragraph with the sentence:

> The manner in which the GERB-like fluxes are used in the GERB processing places no requirements on their absolute accuracy and very limited requirements on their relative accuracy.

And merged this with the following paragraph which has also been slightly altered to accommodate the change.

*17) Figures 6/7 - could these be designed to fit better on the page and maximise the size of the panels? Does the line at 25oS relate to a dead or damaged pixel?*

Answer: The panels have been rearranged to maximise their size as suggested.

The line at around 25 S relates to GERB pixels with a longer than ideal time-constant and that are more sensitive to instrument conditions as a result. For the months shown this can cause an elevation of around $1\mathrm{Wm}^{-2}$ (~0.5%) to the GERB monthly hourly average flux for that region. Discussion of this seemed beyond the scope of the paper as the intent is to investigate the ability to replicate the GERB data. Although noticeable in the ratio because it is a persistent effect, the response difference is quite small and still falls within the overall accuracy statements of the GERB products. Thus, it is not of great relevance to the Obs4MIPs products themselves.

*18) L251 - of course it is feasible to fill missing GERB data with GERB like - there is no need to hypothesise, just quantify the expected error and assess whether this is tolerable for the designed usage of the dataset. A key line to emphasise is that the strategy is to "fill missing GERB fluxes with the corresponding GERB-like fluxes, adjusted by the GERB/GERB-like ratio calculated at the monthly hourly mean temporal and 1º spatial scale." and now the associated errors are quantified.*

Answer: Indeed, I suspect that we intended to hypothesize that it would be feasible to produce a more accurate product by using the GERBlike to fill the missing GERB data. In response to this comment and comments 20 and 21 the paragraph has been rewritten to make the point of the investigation shown in figure 8 clear. (see response to 21 for more information).

*19) L259 1 degree "latitude/longitude" (throughout)*

Answer: Amended as suggested throughout.

*20) Figure 8 - make text bigger. It's not obvious what the use of this is since the decision to scale GERB-like with the ratio seems obvious (and the mean bias of about zero is by design)*

Answer: The text size has been increased (for consistency similar changes have also been made to similar figures). With regards to the point of the plot please see the reply to the following comment in reply to point 21.

*21) L270 - the improvement in agreement between GERB and GERB-like after essentially removing the bias is obvious by design. This sentence can be removed. Section 2.5 seems to be the main analysis following the rational for the approach, which could be more concisely presented as the method*

Answer: We apologise that the intent of this figure wasn't clear and agree that it is more appropriately part of the method. This figure is included to show how a monthly correction impacts the match between the GERB and GERB-like daily hourly data, as the latter is the scale at which the corrected GERB-like will be used as a proxy for missing GERB data. Without this check we have not shown that a monthly correction is useful for our case. It is not a given that the monthly correction would improve the daily data. For example, if the overall bias changes from day to day, the monthly correction would not reduce the range in the daily error distribution mean, shown in the upper panels. Although it would be expected as you note to shift their values to fall around zero. Similarly, if the bias for different scenes occurring at a given location was very different, the monthly average scene bias would not provide a good location specific correction and the standard deviation of the daily error distributions would not be reduced after the monthly correction.

We appreciate that this point was not clear in the original description and the paragraph has been reworked. Also, the whole section is reordered to make clearer the distinction between method and results with this figure forming part of the filling method as you suggested.

*22) Figure 9 - it's not clear to me what the benefit of showing all the unfilled results as well as all the filled results. If the idea is to compare Figure 9 and Figure 4 then an unfilled vs filled line could be presented on the same figure. Are all times lumped together? In this case massive errors due to missing parts of the diurnal cycle will be introduced won't it (which would obviously not be considered in practice)?*

Answer: We have reworked this whole section to make clear that we are discussing the error in the unfilled products due to missing data (original Figure 4) and the filled products (original figure 9) due to filling this missing data. In both cases these error statistics summarize the errors at the 1 degree scale in the **monthly hourly** average. There are no times of day missing, rather this is the effect of some days missing at a given hour and, for the filled case, result in figure 9 being replaced with the adjusted GERB like.

To clarify we have added this detail into the sentence:

"Comparing the two figures shows that filling the missing GERB fluxes with their scaled GERB-like equivalents reduces both the mean error and the spread in the error by more than a factor of 10 in all cases "

So it now reads:

Comparing the two figures shows that filling the missing days of GERB fluxes with their scaled GERB-like equivalents before calculating the monthly hourly average, reduces both the mean and standard deviation of the error in the monthly hourly average at the 1 degree scale by more than a factor of 10 in all cases.

In response to reviewer 2 we have also reordered and reworked this whole section so that it now clearly separates the methodology for producing unfilled and filled products from the evaluation of the products. This clarifies that the two figures are making distinct points relevant to the points in the paper they are included. It is also relevant to compare the figures to demonstrate the value of filling to produce a higher fidelity average. However, as the scales used in figure 4 and 9 are very different (due to the factor of 10 reduction in error), including the results again in figure 9 for comparison would make the results of interest here

(the error after filling) difficult to see well. (Note figure numbers in the updated paper are now figures 9 (previously 4) and 10 (previously 9))

*23) L306 AMIP needs to be explained*

Answer: Now defined with an added reference

*24) Section 3.1 - although important and well described, this is a bit of a gear change from describing the dataset. Is all this information necessary?*

Answer: We have now simplified and summarised the detail on the model differences emphasising the expected changes relevant to our analysis. We have also added a sentence at the end of the introduction to make clear the intent and scope of this section.

*25) (a) Figure 10/11 are very qualitative and it is rather difficult to link colours on the bar to values in the plots. A plot of differences would be more informative.*

Answer: These figures are intended to show a spatially resolved picture that is familiar for orientation to indicate the nature of the information in the GERB product and to highlight regions of interest for more quantitative analysis. They are not expected to be the focus of detailed quantitative analysis. We have added a couple of sentences to make this intent clear at the point they are introduced. In addition to help with linking the colour scale to the contours we have extended the range of colours used for the figures and added two labelled contours on each plot to help orientate readers. Viz:

[Figure]

Due to differences between the model and GERB obs4MIPs product scales (1.875° longitude by 1.25° latitude for the model and 1 by 1 degree for the GERB) a subtraction would necessitate a rescaling and be counter to the intent here which is to illustrate the GERB product directly. We also feel that the differences would be harder to interpret, due to the effects of shifts in regimes, and less useful for orientation than the more familiar monthly average. Illustrative quantitative comparisons which are averaged over larger regions to minimise effects of the different product scales and allow for slight shifts in regime location follow later for the selected regions of study.

*25) (b) Why are coincident model years not used (CMIP6 amip simulations usually end in 2014)?*

Answer: Whilst coincident years were available for GC3.1 runs, only years between 2000 to 2009 were available for the GC5.0 runs. We considered the priority to match model years and use sufficient years to average internal variability as much as possible, as the focus was on comparing general model behaviour with the general behaviour seen in the data. We did consider comparison only for the overlapping years for the three datasets (i.e. 2007, 2008 and 2009), as noted in the paper the average region results for these years only, differ by no more than 3 Wm-2 from the full 2000 to 2009 average. Although not discussed in the paper we have compared for GC3.1 the decade used with 2007-2012 average which matches the GERB

observation period. The differences in this case are even smaller and do not change the conclusions drawn.

*25 (c) I am not sure the Figure 12 results are very relevant to the GERB data description; a reference could suffice.*

Answer: The intent of this figure is to provide detail on the general/average cloud property changes between the two models, to provide context to the flux comparison against the GERB data. It also serves to demonstrate the additional utility of diurnally resolved flux comparisons compared to diurnally averaged or single time of day cloud property comparisons.

*25) (d) Figure 13/14 seem much more relevant - a legend to denote model simulation is needed and perhaps a thick mean (or median) model value would be useful. Showing albedo may reveal the diurnal cycle of stratocumulus better than RSW (which is more dependent on the insolation). Figure 14 - if the idea is to compare the model versions, it would be more useful to have the two mean lines (and perhaps shading for range) in the same plot.*

We have updated these figures, adding the model simulation lines to the legend and including a line for the model mean in each panel as suggested. However, rather than show a model spread we have retained the individual year lines as we feel that the individual annual shapes and variability are more informative.

The GERB Obs4MIPs are monthly hourly mean flux products. Conversion of the monthly hourly mean flux to an albedo would require a deconvolution to account for the daily variation of the of the outgoing flux and incoming solar, this is beyond the scope of the monthly diurnal average product presented. We also note that the GERB RSW are directly linked to the observed radiance which is converted to flux via directional models and not linked to any assumption regarding the incoming solar. Thus, there is an argument that the GERB flux is the more direct observed quantity and for that reason the better quantity to compare against the model.

*26) L352 - it could be mentioned that these issues are not fully solved in higher resolution simulations e.g. Watters et al. (2021) J. Clim doi: 10.1175/JCLI-D-20-0966.1.*

Answer: A sentence noting this has been added including the suggested reference.

*27) Table 1 - how much is positional/regime error and how much is cloud property error? How have biases changed since earlier analysis (e.g. for the NWP version comparison in Allan et al. (2007) QJRMetS clouds were instead too bright due to too much water)?*

Answer: We don't have a direct comparison against the NWP model configuration used in Allan at al. (2007), but the "too few and too bright" bias in low-level clouds has been a persistent problem for many generations of models and is still present in GC3.1 and in other CMIP6 models (Konsta et al., GRL, 2022; https://agupubs.onlinelibrary.wiley.com/doi/10.1029/2021GL097593). It should also be borne in mind that Allan et al 2007 predates the recommendation to apply aging corrections to the GERB RSW. For the GERB 2 July 2006 data used in that study this correction would result

in around a 2% uplift to the GERB fluxes (assuming that the 0.976 correction recommended for the GERB 2 data was available and applied in the Allan comparison).

For our comparisons, altering the region or month under consideration changes the details of the numbers, but not the overall pattern seen. For convection for example, using the African land region from Allan et al. 2007, for either June or July, reduces the SW and increases the OLR compared to the June deep convective region used here. However, the basic result that the GERB data is brighter (by ~20Wm-2) and colder (by ~20-30 Wm-2) than the models remains. With CG5.0 showing a small movement towards the GERB observations.

Regarding the effect of regime position differences, as the region comparisons average monthly averages over a large geographical area and consider multiple years, they comprise both cloud and non-cloud regimes to some extent in all cases and are relatively insensitive to small regimes shifts. The spatially resolved averages for the regions chosen show that the regimes are contained within the chosen geographical areas for both models and observations. However, for the models, as the convection is less extensive, the regime will comprise a smaller portion of the region. You might consider this a positional difference, as a comparison that was limited to identified convection would differ. Although the fact that the convective region were smaller / less intense in the model would remain, which can be considered a cloud property problem, i.e. a lack of cloud development.

The cloud fraction and optical depth plots (original Figure 12) shows that average cloud properties have indeed changed but the effect on the average fluxes (table 1) is small in comparison to the difference between model and data. As the GERB product does not provide associated cloud property information, cloud property effects are not assessed here, but could be the subject of a future study.

The primary intent of the presented study is to compare the diurnal cycle of flux and we feel it achieves this goal. The regions chosen cover the intended regimes for both models and observations. Thus, the flux differences for the regions reflect the differences associated with the regime, but is a function of both regime extent and cloud properties.

*28) L442 the model seems to overestimate OLR for both months - does cirrus outflow or water vapour contribute?*

Though this would be interesting to determine in a future comparison it is beyond the scope of the paper and would likely require higher resolution data and additional information to investigate. We do not discuss it in the paper but in general water vapour comparisons between the model and reanalysis look OK. We cannot say anything definitive about cirrus outflow. As we note in the paper, our comparison between observations and model shows that for the model convection is too weak or missing. There are indications from the monthly mean that the region of deep convection-like low OLR is smaller in the models. The diurnal cycle also shows that the OLR in the models is too warm before the onset of the convection and the convective development is stunted compared to the observations.

*29) L475 - it is still not clear what this factor of 10 reduction in uncertainty means. Is this just for simplistic averaging, where parts of the diurnal cycle are missing and which would not be undertaken in a serious analysis? Or is it referring to missing days?*

Answer: The errors being discussed are at the scale of the product (1 degree longitude/latitude monthly hourly average) no further averaging over the diurnal cycle is being undertaken. The reworking of the original section 2 now makes clear there is an unfilled and filled product, this should make it clearer that we are comparing the errors for a given amount of missing days between the filled and unfilled products. The error in the unfilled GERB Obs4MIPs products due to missing data arises purely from having missing days of data for a given hour when we make the monthly average for that hour and is a result of uncaptured day to day variation. The error in the filled GERB Obs4MIPs products due to filling the missing days is due to the residual difference between the corrected GERB-like and the GERB data it represents. We have also added some words here to clearly define the errors we talking about:

> For a given number of missing days the residual uncertainty in the monthly hourly average at the 1 by 1 degree longitude/latitude scale due to filling is more than a factor of 10 smaller than the error in the unfilled products due to the missing data.

*30) L478 Some clear statements about product uncertainty and it's recommended use could strengthen the conclusions.*

Answer: Additions have been made as suggested.

*31) L479 some illustrative comparison with model simulations are useful for introducing the dataset though I think the work presented here is more deserving of a separate, more detailed investigation.*

Answer: We agree that an expansion of the comparison should be the subject of a more detailed investigation but feel that the comparisons presented in the paper are appropriate to the introduction intended. We feel that both the brief spatial comparison and regime specific diurnal investigation provide context and show the unique strength of the product. The presented studies also clearly illustrate how non-diurnally resolved cloud property comparisons are complementary but are, on their own, insufficient to understand improvements in cloud response on diurnal timescales.

*32) Acknowledgements - the work of the GERB team could be mentioned in the acknowledgments and more specifically the contribution to initial discussions of the strategy for dealing with sunglint led by Jo Futyan and the terminator led by me could be stated.*

Answer: These were not included because they pertained to the GERB data where they have long been implemented and are not specific to the Obs4MIPs products presented here. However, we agree that these features of the GERB product are fundamental to the ability to produce an average and thus along with the availability of the GERBlike data are of fundamental importance to the obs4MIPs product. We have thus added acknowledgements as suggested as well as to the RMIB team for their help with the GERBlike dataset

---

## Author Comment (AC2)

**Reply to reviewer 2.**

We thank the reviewer for their time and helpful suggestions, particularly as regards to the suggested rearrangement of section 2 into methods and results, which we agree improves the clarity of the paper.

In respect of their overarching comment:

*First, they should clarify how their dataset improves and complements the existing TOA satellite-based products. A comparison of GERB Obs4MIP against state-of-the-art products (e.g., CERES) would be highly recommended. Second, they should provide a clearer description of the methodology, splitting between methodology and results, so manuscript readers and dataset users can find more easily all the methodological steps used to produce the dataset.*

We give more detailed replies in the response to the detailed comments below, but to summarize here: the revised paper adds a discussion of the relationship between the GERB data and other observational top of the atmosphere flux products and has provided a clearer split between methodology and results as suggested. We have also added references to previous comparisons between GERB and CERES products and explain below why a further comparison between the GERB Obs4MIP and temporally interpolated CERES products does not provide any additional benefit.

We provide detailed replies to the specific comments below which we have numbered for ease of reference.

**Major comments**

*1)-Section 1 – Introduction. I would suggest including a summary of the existing satellite-based products for RSW and OLR (e.g., CERES, NASA GEWES SRB, CLARA). It is true that most of the existing products are based on polar-orbiting instruments that focus on daily and monthly fluxes. However, products such as CERES SYN and NASA GEWEX SRB are also able to represent the diurnal TOA cycle (3-hourly resolution). The authors should highlight how their GERB Obs4MIP can complement and improve these products.*

Answer: As suggested more detail has been added to section 1 to mention other satellite measurements of the TOA RSW and OLR. Specific mention has been made of CERES, ERBE and ScaRAB datasets. However, we do not discuss the NASA GEWEX SRB as this is a surface not TOA product. We also do not consider the CLARA dataset as this is not a satellite based *observation* of the RSW and OLR, but a calculation of these quantities based on satellite retrieved quantities about the state of the atmosphere, surface and cloud properties. In this sense it is more of a reanalysis product, and we don't think it is relevant to include in a discussion about observations of the OLR and RSW unless one were to extend the discussion to include reanalyses as well.

We note that the CERES instrument is polar orbiting and thus the diurnally resolved product relies on narrowband observations to supplement the baseline observations. This means that the CERES diurnally resolved products are therefore somewhat different in nature to the

GERB data presented here, as the diurnally resolved component is not based on broadband observations. A discussion of this, including references to the use of the GERB products in the development and evaluation of the CERES temporal interpolation that underlies the CERES higher temporal resolution products, has been added.

*2) -Line 73 - temporal averaging: Why are the temporal averages not weighted as done for the spatial averages? The 15-minute instantaneous GERB observations are not perfectly aligned with the hourly UTC intervals. Moreover, the exact timestamp of the retrieval changes between pixels following the GERB scanning cycle. First, this temporal mismatch should be described in the manuscript. Second, due to this mismatch, authors should consider 5 GERB observations (instead of 4) and weight the observations at the edge of the hourly interval accordingly.*

Answer: The original GERB observations are indeed not perfectly aligned with the hour and differ between columns. However, the GERB HR product used here has already undertaken the interpolation of the original GERB observations to provide a product which is a snapshot at the 15 minute SEVIRI observation time: this interpolation does indeed use GERB observation beyond the original hour. As this is part of the production of the GERB HR product and described in detail in the reference given when we introduce the use of the GERB HR, we don't feel it is appropriate to discuss in detail in this paper. However, we have added an additional sentence at the start of section 2.1 to emphasise that the HR product is a temporal interpolation of the original GERB observations:

> The GERB HR fluxes are a temporally interpolated resolution enhanced version of original GERB observations, derived using spatial information on the scene variation within the GERB footprint from the SEVIRI imager.

The GERB HR product replicates the SEVIRI time difference according to row, which is the result of the SEVIRI scan mechanism. The products are named according to the time of the start of the scan and run from this time for the top (northernmost) row, to 12 minutes later for the bottom (southernmost) row. Using the four HR products starting at 00, 15, 30 and 45 past the hour we ensure that all the observations from all rows have been observed during the 60 minutes of the given hour. As stated in the methods, in deriving the hourly averages for the shortwave we average albedo, essentially adjusting for the variation in incoming solar to the centre of the hour for all pixels. We have updated the paper to note that the albedo averaging and sza adjustment to the central value, mitigates not just for missing timeslots but also for the slight time offset that occurs with row. No adjustment for the variation of albedo with solar zenith or for the scene variation with time is attempted. This level of complexity is unlikely to add further benefit for this small time offset considering the likely uncertainty that would be added by any attempt to adjust these. The choice to present the RSW flux for the hour as the average albedo multiplied by the incoming solar at the centre of the 1 degree region and hour was decided during product development at the request of modellers as being most appropriate for model comparison.

*3) -Line 75 - temporal averaging: explain better the albedo conversion. I assume that incoming solar radiation is calculated twice. First, at the GERB retrieval timestamp (to transform GERB instantaneous RSW observations into instantaneous albedo), and then, at the center of the UTC hourly interval (to transform hourly albedo averages into hourly RSW averages). I would suggest including the corresponding equations to clarify this part of the methodology.*

Answer: This is correct but the only difference between the incoming solar calculated is the change in solar zenith angle. We have reworked this explanation to clarify that the process is really an adjustment of the solar zenith, as other terms cancel, showing the ratio used for the adjustment:

> As the total solar irradiance and the Earth-Sun distance do not change during the conversion to albedo, and back to flux, this becomes purely an adjustment in solar zenith angle to the centre of the grid box and hour bin. The process is equivalent to multiplying each flux by the ratio $\cos(\theta_{local})$ / $\cos(\theta_{centre})$, where $\theta_{local}$ is solar zenith angle at the HR pixel time and position and $\theta_{centre}$ is the solar zenith angle at the 1 degree latitude/longitude centre at half past the hour.

*4) -Line 75 – spatial averages: Why was the albedo conversion not applied to the spatial averages? The same bias mentioned in line 77 for the temporal average could be introduced in the temporal averages if RSW instantaneous observations are systematically missing at some parts of the 1x1 degree pixel, due to the change of RSW with latitude.*

Answer: The conversion to albedo is applied before *any* averaging (both spatial and temporal), and the conversion made back to flux at the 1 degree hourly scale. We have amended the sentence to explicitly state that the conversion is done before both spatial and temporal averaging.

*5)-(missing) methods section. The current version of the manuscript presents a sequential structure that mixes methodology paragraphs with results paragraphs. For instance, the averaging and gap-filling processes are currently described in two different sections (section 2.1 and section 2.4), while sections 2.2 and 2.3 contain methods and results regarding the impact of missing data. I consider that the readability of the manuscript could improve by splitting methods and results. The methods section could be further split into "dataset production" (a kind of ATBD) and "dataset evaluation" (e.g., the impact of missing data before and after gap-filling). A specific section on the "dataset production" containing all the methodological steps (including an extended version of Fig 1 diagram, which currently only focuses on the averaging process) would be highly valuable for potential users.*

Answer: We thank the reviewer for this excellent observation and agree their suggested change would be much more helpful to users than the current layout which describes a first attempt at an unfilled dataset and then justifies the need to improve it. We have rearranged these sections in accordance with the suggestion. We now describe the production of an unfilled and filled dataset (both are released Obs4MIPs GERB products) and then present the evaluation of each.

*6)-I would also suggest adding a specific section or sub-section listing all the attributes of the final product (e.g., spatial and temporal resolution, spatial and temporal coverage, data format, data layers available in the final product, etc.).*

We have added this information into the data availability section and also added a summary in the conclusions.

*7)-Dataset evaluation: The manuscript would significantly improve if the authors included a validation of their dataset against an external TOA satellite-based product. The obvious choice would be using CERES products. This will not only allow benchmarking the new*

*dataset against state-of-the-art products but also having an independent reference to quantify the improvement obtained with some of the methodological steps proposed by the authors (gap-filling, GERB/GERB-like ratios)*

Answer: The original GERB data has already been compared with the CERES products for observationally matched points (i.e. GERB points matched to direct CERES observations). This is a fundamental comparison and evaluation of the GERB observational dataset against the CERES observational dataset (Clerbaux et al 2009, doi:10.106/j.rse.2008.08.016, Parfitt et a 2016, doi: 10.1016/j.rse.2016.09.005, 2016). Furthermore, the fidelity of the CERES temporal interpolation used to produce their SYN product has also been evaluated using observations from GERB (Doelling, et al. 2013 doi: 10.1175/JTECH-D-12-00136.1 & 2016 doi: 10.1175/JTECH-D-15-0147.1). All these references have now been included in the paper.

These, much more direct, comparisons exploit the strengths of both datasets: CERES for baseline accuracy, GERB for a correct (observed) representation of the diurnal variation in outgoing fluxes. We argue that comparing the two hourly monthly mean products would essentially mimic these studies but with the complication of combining accuracy and sampling differences that have already been assessed. Hence we argue against performing this exercise. The evaluation presented in this paper deals specifically with the additional problem of creating an average where there is incomplete data which is the only additional consideration in evaluating the fidelity of the GERB Obs4MIPs monthly hourly averages.

**Minor comments**

8) -Line 84: "Hence, twilight and night-time RSW HR fluxes are not included in the averaging to the daily hourly scale if the central solar zenith angle is less than 85 but are used to replace grid-box values when the central solar zenith angle is equal to or exceeds 85" Could you clarify this sentence? Regarding the first part of the sentence, does it mean that 9km pixels with SZA > 85 are not used in the spatial average if the SZA at the center of the 1x1 pixel is less than 85 degrees?

Answer: Correct, if the SZA at the centre of the hour for that 1 degree region is less than 85 degrees the flux value of any GERB HR pixels with SZA > 85 are not included in the spatial or temporal averaging. This is because these pixels are not observations but just a fixed non location dependent twilight model flux used to avoid bias when diurnally averaging and would not be appropriate to include except for a twilight location. By excluding these twilight model values completely when the central point has a SZA < 85° and using only these values when the SZA ≥ 85° we preserve the diurnal bias correction and avoid them corrupting observed values. We have stated more clearly that they are excluded from the spatial and temporal averaging and noted that they are not observed quantities.

*9)-Line 194: "empirical narrowband to broadband conversion" Please, include either the equation with the coefficients or a reference to a document describing this conversion.*

Answer: References have been added as requested.

*10)-Figures 2 & 5: Is there any reason to use these unevenly spaced categories (0, 1-5, 6-22, >22) for the number of missing days? If so, explain it. Otherwise, I would suggest using an evenly-spaced color palette or a continuous color palette*

Answer: The colour coding is related to the data included in the filled and unfilled products discussed later, with up to 5 missing days being allowed in the released unfilled products and up to 22 filled days being allowed in the released filled products. No missing days are highlighted separately to show the cases not subject to error due to missing or filled data. We have added information regarding this to the figure caption, although detailed discussion is left until later in the paper.

*11)-Figures 3, 4, 6, 7, 8, 9: The panel number is not seen very well. Please, take it out of the panel and increase the font size (and/or use bold text).*

Answer: Panel numbers have been moved to the outside of the panels and font size increased as suggested.

*12)-Figure 3: I would suggest using a diverging color palette centered around 0. Otherwise, it is difficult to interpret the differences. I would also suggest describing the four realizations in the figure caption.*

Answer: We apologise, an incorrect key was provided on this figure, a diverging colour scale was in the plot but not shown in the key. This has been corrected and a diverging colour scale is used and shown in the colour bar.

*13)-Figure 4 & 8. Add a horizontal line (in the background) to better interpret the bias plots.*

Answer: A horizontal zero line has been added to the error distribution mean panels as suggested

*14)-Figure 6 & 7. As for Figure 3, use a diverging color palette centered around 1*

Answer: We chose a linear colour scale for these plots because it is the spatial and temporal variation in the ratio rather than its difference from 1 that is most significant. A fixed offset would be a simple adjustment but the variability of the difference in time and space requires more complex treatment. A diverging colour scale centred around 1, although sensible in terms of 1 being an important value, would not be useful given the distribution of the ratios. For the RSW less than 1% of the points have a ratio less than 1, and for the OLR less than 2.5% of the points have a ratio greater than 1. Thus, using a colour scale centred around 1 for these plots would limit the plotted colours to around half of those available and would barely make use of the diverging nature of the scale. We show some examples below for the RSW which illustrate this issue:

[Figure]

As an alternative, a diverging colour scale centred around the mean bias (1.076 for the SW and 0.99 for the OLR) could be used. However, there is nothing particularly special about the mean, either physically, or in relation to scene or angle. Thus, we feel this choice gives unwanted emphasis to the contrast between points above and below the mean even when they are quite close in value, at the expense of emphasizing the overall range of values which is more relevant to consider. An example using a diverging colour scale centred around the mean for RSW is shown below to illustrate this.

[Figure]

For these reasons we feel a diverging colour scale centred around 1 or the mean would be counterproductive to what these plots are trying to explore. We have added some text to the discussion of these plots to emphasise that it is variability in the ratio rather than its deviation from 1 that we are most interested in considering when determining the nature of the correction required.

*15)-Section 2.5: Could you clarify if GERB-like fluxes are used (a) only to replace fully missing 1x1 degree averages, or (b) also to replace missing 9km GERB HR observations before the spatial averaging?*

The corrected GERB-like fluxes are only used to replace fully missing 1x1 degree averages. This has now been more clearly explained and the following schematic added to clarify the filling process:

[Figure]

.

*16)-Discuss the challenges to extend this methodology to GERB instruments onboard other MSG satellites, and if you have any plans to undertake this project in the near future.*

A discussion of this has been added to the conclusions as suggested.

---

## Author Response (AR1)

**Authors Response**

**Point by point response**

**Response to points raised by reviewer 1**

*"Overall, my opinion is that a more focused description of the data, highlighting strengths and weaknesses and presenting quantitative assessment of the uncertainty would improve the impact of this dataset. Although interesting, I am not sure the illustration of the dataset's use in evaluating a climate model simulation is central to a dataset description paper and would benefit from a more detailed exploration elsewhere. However, if the authors prefer this more lengthy discussion, I do not see any scientific reason to object and I consider that this work is suitable for publication with only minor modifications listed below."*

We think that the changes made to what was section 2, into a data production (new **section 2**) and data evaluation (new **section 3**) sections, more clearly emphasises the assessment of the dataset. We prefer to keep the inclusion of the brief comparison between model and data to highlight the utility of the dataset and demonstrate the information it can provide to study diurnally varying processes, such as the examples used. To this end we focus on the comparison of the diurnal cycle of radiation in two regions showing how this provides unique information that can be not gleaned from looking at cloud property changes alone. We agree that further papers could certainly explore this further we now note this in the conclusion (**lines 551 to 2**) but would not wish to remove the discussion included in this paper nor significantly expand it by introducing other observational datasets.

We provide answers to the specific questions below.

*1) Title - I would not describe GERB Obs4MIP as a tool? How about something like: "The GERB: a dataset for evaluating diurnal and monthly variation in top of atmosphere radiative fluxes in climate models."*

Answer: Title has been changed as suggested. **Lines 1-2**

*2) L8 - since GERB data has been used in model evaluation for 20 years it is not new. What is new is the Obs4MIP aspect ("newly reprocessed" may be more appropriate).*

Answer: We have changed 'new' to 'newly available'. We choose not to use 'newly reprocessed' to avoid any implication that the underlying processing of the GERB data has changed. The GERB Obs4MIPs data presented are newly derived higher level products based on the long existing processing of the GERB data. **Line 8.**

*3) L11 "approximately" since it is not a square 60S-60N, 60W-60W domain*

Answer: Added as suggested. **Line 11**

*4) L19 ":" --> "," (or we could compromise with a ";")!*

Answer: Altered to a comma as suggested **line 20.**

*5) L20 improved fidelity, relative to GC3.1*

Answer: Amended as suggested **line 21**

*6) L27 Since outgoing longwave is used later it may be worth stating here e.g. "emitted thermal infrared (outgoing longwave) and solar reflected..."*

Answer: Amended as suggested noting the alterations in section in respect of comments from reviewer 2 **line 29.**

*7) L36 also Allan et al. (2011), Examination of long-wave radiative bias in general circulation models over North Africa during May–July. Q.J.R. Meteorol. Soc., 137: 1179-1192. https://doi.org/10.1002/qj.717*

Answer: Added as suggested. **Line 50**

*8) L55 add a line pointing to the fact that the HR product is a resolution enhanced version of the GERB broadband radiative fluxes using SEVIRI narrow band measurements.*

Answer: As suggested the following sentence has been added **lines78 to 80.**

*9) L63 "are" --> "were"*

Answer: Altered as suggested **line 80.**

*10) L72 why is 70 degrees zenith the cut-off and not slightly more or less?*

Answer: The sentence has been amended to indicate that the vza limit relates to the recommended valid range of the GERB products when averaging the data. **Line 96.**

*11) L74 "proceed without prejudice" is not really clear what is meant (sounds like legal speak). Is one observation representative of a 100x100km area? Presumably there are mostly lots of values per 1x1 degree box.*

Answer: Modified to clarify (**Lines 98 to 101).**

*12) L76 incoming solar flux from what dataset or algorithm?*

Answer: **lines 104 to 108** now clarify that this is in effect a multiplication by the solar zenith angle ratio for the original HR to that at the 1 degree centre and there is no dependence on an assumed incoming solar beyond this multiplier.

*13) Figure 3 - the colour bar seems to bear no resemblance to the maps. Also this figure could be designed in a way to maximise the size of the panels (or reduce the size of the plot and all the dead space)*

Answer: Colour bar corrected and figure altered as suggested Our apologies an incorrect scale was included. **Figure 8**

*14) L187 "exiting" --> "existing"*

Answer: Amended as suggested. **Line 155**

*15) L200 this paragraph could be reduced to the first line plus "(compare Figures 5 and 3)".*

Answer: We have reduced to 3 sentences **Lines 166 to 169.**

*16) L205 these lines are difficult to understand. Can they be written more clearly? There is a lot of detail in this section that may be unnecessary for users of the dataset so another option is to more briefly note issues and refer to prior papers if the reader is interested.*

Answer: We have replaced this paragraph removing the unnecessary detail. **Lines 170 to 171**

*17) Figures 6/7 - could these be designed to fit better on the page and maximise the size of the panels? Does the line at 25oS relate to a dead or damaged pixel?*

Answer: The panels have been rearranged to maximise their size as suggested. **Figures 4 and 5.** The line at around 25 S relates to GERB pixels with a longer than ideal time-constant and that are more sensitive to instrument conditions as a result. For the months shown this can cause an elevation of around $1Wm_{-2}$ (~0.5%) to the GERB monthly hourly average flux for that region. Discussion of this seemed beyond the scope of the paper as the intent is to investigate the ability to replicate the GERB data. Although noticeable in the ratio because it is a persistent effect, the response difference is quite small and still falls within the overall accuracy statements of the GERB products. Thus, it is not of great relevance to the Obs4MIPs products themselves.

*18) L251 - of course it is feasible to fill missing GERB data with GERB like - there is no need to hypothesise, just quantify the expected error and assess whether this is tolerable for the designed usage of the dataset. A key line to emphasise is that the strategy is to "fill missing GERB fluxes with the corresponding GERB-like fluxes, adjusted by the GERB/GERB-like ratio calculated at the monthly hourly mean temporal and 1° spatial scale." and now the associated errors are quantified.*

Answer: In response to this comment and comments 20 and 21 the paragraph has been rewritten to make the point of the investigation shown in figure 8 clear. (see response to 21 for more information). **Lines 210 to 212** replaces a discussion of hypothesis and clarifies that we need to consider how well a monthly correction adjusts the daily data that will be used for filling.

*19) L259 1 degree "latitude/longitude" (throughout)*

Answer: Amended as suggested throughout the paper.

*20) Figure 8 - make text bigger. It's not obvious what the use of this is since the decision to scale GERB-like with the ratio seems obvious (and the mean bias of about zero is by design)*

Answer: The text size has been increased (for consistency similar changes have also been made to similar figures). **Figures 6, 9 and 10.** The point of the plot is to consider how well a monthly

ratio can correct the daily differences implicitly determining the day to day variation in the required correction. **Lines 216 to 224** now explain this**.** See also reply to point 21.

*21) L270 - the improvement in agreement between GERB and GERB-like after essentially removing the bias is obvious by design. This sentence can be removed. Section 2.5 seems to be the main analysis following the rational for the approach, which could be more concisely presented as the method*

Answer: This discussion is now included in a 'methods' section describing the production of the unfilled and filled products (**Section 2**). Here we are considering differences between the GERB and GERB-like at the *daily* hourly scale after adjustment by the *monthly* hourly ratio. The relationship between optimal daily and monthly corrections depends on the day to day variation of the differences which we have not yet considered. Daily variation in overall bias or different corrections for different scenes at a given location impact the ability of the monthly correction to improve the daily data. Without this check we have not shown that a monthly correction is useful for our case as it is the *daily* hourly corrected GERB-like that will be used as a proxy for missing GERB data. **Lines 217 to 224** now clarify this.

*22) Figure 9 - it's not clear to me what the benefit of showing all the unfilled results as well as all the filled results. If the idea is to compare Figure 9 and Figure 4 then an unfilled vs filled line could be presented on the same figure. Are all times lumped together? In this case massive errors due to missing parts of the diurnal cycle will be introduced won't it (which would obviously not be considered in practice)?*

Answer: Both figures are now presented in a reworked 'results' type section described the evaluation of both the unfilled and filled products (**Section 3**) making the purpose of the plots clearer. The figures show the error in the unfilled products due to missing data (**Figure 9**) and the filled products (**Figure 10**) due to filling this missing data. In both cases these error statistics summarize the errors at the 1 degree scale in the *monthly hourly* average this is stated it the figure captions (**lines 315 and 347**) and the discussion (**lines 300 and 335**). There are no times of day missing, rather this is the effect of some days missing at a given hour and, for the filled case being replaced with the adjusted GERB like. The comparison of the figures is clarified **Lines 336 to 339** The two figures are making distinct points relevant to the points in the paper they are included. They are compared to demonstrate the value of filling to produce a higher fidelity average. However, as the scales used in the two figures are very different (due to the factor of 10 reduction in error), including both results in a single plot would make the detail difficult to see for the filled error.

*23) L306 AMIP needs to be explained*

Answer: Now defined with an added reference. **Line 356.**

*24) Section 3.1 - although important and well described, this is a bit of a gear change from describing the dataset. Is all this information necessary?*

Answer: We have simplified and summarised the detail on the model differences emphasising the expected changes relevant to our analysis (**lines 366 to 372**). We have also added a sentence at the end of the introduction to make clear the intent and scope of this section (**lines 358 to 359**).

*25) (a) Figure 10/11 are very qualitative and it is rather difficult to link colours on the bar to values in the plots. A plot of differences would be more informative.*

Answer: To help with linking the colour scale to the contours we have extended the range of colours used for the figures and added two labelled contours on each plot to help orientate readers **Figures 11 and 12**. These figures are intended to show a spatially resolved picture that is familiar for orientation to indicate the nature of the information in the GERB product and to highlight regions of interest for more quantitative analysis. They are not expected to be the focus of detailed quantitative analysis. We have added a couple of sentences to make this intent clear at the point they are introduced **lines 400 to 401**.

Due to a mis match between the model and GERB Obs4MIPs spatial resolution ($1.875_o$ longitude by $1.25_o$ latitude for the model and 1 by 1 degree for the GERB) a subtraction would necessitate a rescaling and be counter to the intent here which is to illustrate the GERB product directly. We

also feel that the differences would be harder to interpret, due to the effects of shifts in regimes, and less useful for orientation than the more familiar monthly average. Quantitative comparisons which are averaged over larger regions to minimise effects of the different product scales and allow for slight shifts in regime location follow later for the selected regions of study.

*25) (b) Why are coincident model years not used (CMIP6 amip simulations usually end in 2014)?*
Answer: Whilst coincident years were available for GC3.1 runs, only years between 2000 to 2009 were available for the GC5.0 runs. We considered the priority to match model years and use sufficient years to average internal variability as much as possible, as the focus was on comparing general model behaviour with the general behaviour seen in the data. We did consider comparison only for the overlapping years for the three datasets (i.e. 2007, 2008 and 2009), as noted in the paper the average region results for these years only, differ by no more than 3 Wm-2 from the full 2000 to 2009 average (**lines 423 to 425**). Although not discussed in the paper we have compared for GC3.1 the decade used with 2007-2012 average which matches the GERB observation period. The differences in this case are even smaller and do not change the conclusions drawn.

*25 (c) I am not sure the Figure 12 results are very relevant to the GERB data description; a reference could suffice.*
Answer: The intent of this figure is to provide detail on the general/average cloud property changes between the two models, to provide context to the flux comparison against the GERB data (**line 439**). It also serves to demonstrate the additional utility of diurnally resolved flux comparisons compared to diurnally averaged or single time of day cloud property comparisons highlighting that the latter does not inform of improvements to the diurnal cycle flux response.

*25) (d) Figure 13/14 seem much more relevant - a legend to denote model simulation is needed and perhaps a thick mean (or median) model value would be useful. Showing albedo may reveal the diurnal cycle of stratocumulus better than RSW (which is more dependent on the insolation). Figure 14 - if the idea is to compare the model versions, it would be more useful to have the two mean lines (and perhaps shading for range) in the same plot.*
The model simulation lines have been added to the legend and a line for the model mean has been added in each panel as suggested (**Figure 14 and 15**). However, rather than show a model spread we have retained the individual year lines as we feel that the individual annual shapes and variability are more informative as the diurnal variation can get lost if just the mean and spread are shown. Albedo comparisons are not feasible or considered desirable. The GERB Obs4MIPs are monthly hourly mean flux products. Conversion of the monthly hourly mean flux to an albedo would require a deconvolution to account for the daily variation of the of the outgoing flux and incoming solar, this is beyond the scope of the monthly diurnal average product presented. We also note that the GERB RSW are directly linked to the observed radiance which is converted to flux via (empirical) directional models and not linked to any assumption regarding the incoming solar. Thus, there is an argument that the GERB flux is the more direct observed quantity and for that reason the better quantity to compare against the model.

*26) L352 - it could be mentioned that these issues are not fully solved in higher resolution simulations e.g. Watters et al. (2021) J. Clim doi: 10.1175/JCLI-D-20-0966.1.*
Answer: A sentence noting this has been added including the suggested reference. (**Lines 396 to 397**)

*27) Table 1 - how much is positional/regime error and how much is cloud property error? How have biases changed since earlier analysis (e.g. for the NWP version comparison in Allan et al. (2007) QJRMetS clouds were instead too bright due to too much water)?*
Answer: We don't have a direct comparison against the NWP model configuration used in Allan at al. (2007), and differences in the version of the GERB data used in that study needs to be considered. However, the "too few and too bright" bias in low-level clouds has been a persistent problem for many generations of models and is still present in GC3.1 and in other CMIP6 models (Konsta et al., GRL, 2022; https://agupubs.onlinelibrary.wiley.com/doi/10.1029/2021GL097593). Allan et al 2007 predates the recommendation to apply aging corrections to the GERB RSW. For

the GERB 2 July 2006 data used in that study this correction would result in around a 2% uplift to the GERB fluxes (assuming that the 0.976 correction recommended for the GERB 2 data was available and applied in the Allan comparison).

For our comparisons, altering the region or month under consideration changes the details of the numbers, but not the overall pattern seen. For convection for example, using the African land region from Allan et al. 2007, for either June or July, reduces the SW and increases the OLR compared to the June deep convective region used here. However, the basic result that the GERB data is brighter (by ~20Wm-2) and colder (by ~20-30 Wm-2) than the models when making a regional comparison as here remains. With CG5.0 showing a small movement towards the GERB observations.

Regarding the effect of regime position differences, as the region comparisons average monthly averages over a large geographical area and consider multiple years, they comprise both cloud and non-cloud regimes to some extent in all cases and are relatively insensitive to small regimes shifts. The spatially resolved averages for the regions chosen show that the regimes are contained within the chosen geographical areas for both models and observations. However, for the models, as the convection is less extensive, the regime will comprise a smaller portion of the region as noted (**line 438**). You might consider this a positional difference, as a comparison that was limited to identified convection would differ. Although the fact that the convective region were smaller / less intense in the model would remain, which can be considered a cloud property problem, i.e. a lack of cloud development.  The cloud fraction and optical depth plots (**Figure 13**) shows that average cloud properties have indeed changed but the effect on the average fluxes (**table 1**) is small in comparison to the difference between model and data. As the GERB product does not provide associated cloud property information, cloud property effects are not assessed here, but could be the subject of a future study.

The primary intent of the presented study is illustrate the utility of the GERB data by comparing the diurnal cycle of flux and we feel it achieves this goal. The regions chosen cover the intended regimes for both models and observations. Thus, the flux differences for the regions reflect the differences associated with the regime, but is a function of both regime extent and cloud properties. The differences are consistent with the observation of missing land convection in the spatial plots (**line 438 and Figure 12**). The improvements in diurnal cycle of the OLR for the deep convection region are perhaps the most interesting point to consider in this comparison (**line 485 to 490**) as their evaluation is specific to the GERB Obs4MIPs products being showcased, despite the remaining issues regarding the overall level of the convection (**lines 490 to 492**) which would be better explored using additional observational datasets.

*28) L442 the model seems to overestimate OLR for both months - does cirrus outflow or water vapour contribute?*

Though this would be interesting to determine in a future comparison it is beyond the scope of the paper and would likely require higher resolution data and additional information to investigate. We do not discuss it in the paper but in general water vapour comparisons between the model and reanalysis look OK. We cannot say anything definitive about cirrus outflow. As we note in the paper, our comparison between observations and model shows that for the model convection is too weak or missing (**line 491**). There are indications from the monthly mean that the region of deep convection-like low OLR is smaller in the models (**line 438**). The diurnal cycle also shows that the OLR in the models is too warm before the onset of the convection and the convective development is stunted compared to the observations (**figure 15 and lines 490 to 492**), but concentrate on examining the diurnal cycle of the flux to illustrate the use of the GERB Obs4MIPs products and the improvements shown there (**lines 485 to 490**).

*29) L475 - it is still not clear what this factor of 10 reduction in uncertainty means. Is this just for simplistic averaging, where parts of the diurnal cycle are missing and which would not be undertaken in a serious analysis? Or is it referring to missing days?*

Answer: The errors being discussed are at the scale of the product (1 degree longitude/latitude monthly hourly average) no further averaging over the diurnal cycle is being undertaken. The

reworking of the original section 2 into section 2 and 3 now makes clear there is an unfilled and filled product, this should make it clearer that we are comparing the errors for a given amount of missing days between the filled and unfilled products. The error in the unfilled GERB Obs4MIPs products due to missing data arises purely from having missing days of data for a given hour when we make the monthly average for that hour and is a result of uncaptured day to day variation. The error in the filled GERB Obs4MIPs products due to filling the missing days is due to the residual difference between the corrected GERB-like and the GERB data it represents. We have also added some words here to clearly define the errors we talking about (**lines 530 to 536**).

*30) L478 Some clear statements about product uncertainty and it's recommended use could strengthen the conclusions.*

Answer: Additions have been made to the conclusions as suggested (**lines 562 to 564**).

*31) L479 some illustrative comparison with model simulations are useful for introducing the dataset though I think the work presented here is more deserving of a separate, more detailed investigation.*

Answer: We agree that an expansion of the comparison should be the subject of a more detailed investigation and have now noted that (**lines 551 to 552**). However, we feel that the comparisons presented in the paper are appropriate to the introduction intended. We feel that both the brief spatial comparison and regime specific diurnal investigation provide context and show the unique strength of the product. The presented studies also clearly illustrate how non-diurnally resolved cloud property comparisons are complementary but are, on their own, insufficient to understand improvements in cloud response on diurnal timescales.

*32) Acknowledgements - the work of the GERB team could be mentioned in the acknowledgments and more specifically the contribution to initial discussions of the strategy for dealing with sunglint led by Jo Futyan and the terminator led by me could be stated.*

Answer: Acknowledgements have been added (**lines 580 to 585**).

**Response to points raised by reviewer 2.**

*First, they should clarify how their dataset improves and complements the existing TOA satellite-based products. A comparison of GERB Obs4MIP against state-of-the-art products (e.g., CERES) would be highly recommended. Second, they should provide a clearer description of the methodology, splitting between methodology and results, so manuscript readers and dataset users can find more easily all the methodological steps used to produce the dataset.*

We give more detailed replies in the response to the detailed comments below, but to summarize here: the revised paper adds a discussion of the relationship between the GERB data and other observational top of the atmosphere flux products (**lines 26 to 35**) and has provided a clearer split between methodology and results as suggested (**Sections 2 and 3**). We have also added references to previous comparisons between GERB and CERES products (**lines 46, 245, 524**) and explain below why a further comparison between the GERB Obs4MIP and temporally interpolated CERES products does not provide any additional benefit.

We provide detailed replies to the specific comments below which we have numbered for ease of reference.

**Major comments**

*1)-Section 1 – Introduction. I would suggest including a summary of the existing satellite-based products for RSW and OLR (e.g., CERES, NASA GEWES SRB, CLARA). It is true that most of the existing products are based on polar-orbiting instruments that focus on daily and monthly fluxes. However, products such as CERES SYN and NASA GEWEX SRB are also able to represent the diurnal TOA cycle (3-hourly resolution). The authors should highlight how their GERB Obs4MIP can complement and improve these products.*

Answer: As suggested more detail has been added to section 1 to mention other satellite measurements of the TOA RSW and OLR (**lines 26 to 35**). Specific mention has been made of CERES, ERBE and ScaRAB datasets. However, we do not discuss the NASA GEWEX SRB as

this is a surface not TOA product. We also do not consider the CLARA dataset as this is not a satellite based *observation* of the RSW and OLR, but a calculation of these quantities based on satellite retrieved quantities about the state of the atmosphere, surface and cloud properties. In this sense it is more of a reanalysis product, and we don't think it is relevant to include in a discussion about observations of the OLR and RSW unless one were to extend the discussion to include reanalyses as well.

We note that the GERB is the only geostationary ERB mission (**lines 34 to 35**) providing diurnally resolved broadband observations in contrast to polar orbiting CERES instruments where diurnal resolution relies on narrowband observations to supplement the baseline observations. This makes the CERES diurnally resolved products somewhat different in nature to the GERB which has been used for their development and evaluation (**lines 43 to 46**).

*2) -Line 73 - temporal averaging: Why are the temporal averages not weighted as done for the spatial averages? The 15-minute instantaneous GERB observations are not perfectly aligned with the hourly UTC intervals. Moreover, the exact timestamp of the retrieval changes between pixels following the GERB scanning cycle. First, this temporal mismatch should be described in the manuscript. Second, due to this mismatch, authors should consider 5 GERB observations (instead of 4) and weight the observations at the edge of the hourly interval accordingly.*

Answer: The GERB HR product used has already undertaken the interpolation of the original GERB observations we have added a summary of this (**lines 79 to 82**) a detailed description is provided in the Brindley and Russell 2017 reference cited (**line 77**).

The GERB HR product replicates the SEVIRI time difference according to row, which is the result of the SEVIRI scan mechanism. The products are named according to the time of the start of the SEVIRI scan and run from this time for the top (northernmost) row, to 12 minutes later for the bottom (southernmost) row. Using the four HR products starting at 00, 15, 30 and 45 past the hour we ensure that all the observations from all rows have been observed during the 60 minutes of the given hour. As stated in the methods, in deriving the hourly averages for the shortwave we average albedo, essentially adjusting for the variation in incoming solar to the centre of the hour for all pixels. We have updated the paper to note that the albedo averaging and sza adjustment to the central value, mitigates not just for missing timeslots but also for the slight time offset that occurs with row (**lines 110 to 111**). No adjustment for the variation of albedo with solar zenith or for the scene variation with time is attempted. This level of complexity is unlikely to add further benefit for this small time offset considering the likely uncertainty that would be added by any attempt to adjust these. The choice to present the RSW flux for the hour as the average albedo multiplied by the incoming solar at the centre of the 1 degree region and hour was decided during product development at the request of modellers as being most appropriate for model comparison.

*3) -Line 75 - temporal averaging: explain better the albedo conversion. I assume that incoming solar radiation is calculated twice. First, at the GERB retrieval timestamp (to transform GERB instantaneous RSW observations into instantaneous albedo), and then, at the center of the UTC hourly interval (to transform hourly albedo averages into hourly RSW averages). I would suggest including the corresponding equations to clarify this part of the methodology.*

Answer: It has been clarified that the flux to albedo to flux conversions amount to an adjustment in solar zenith angle only and the adjustment ratio is detailed (**lines 104 to 108**).

*4) -Line 75 – spatial averages: Why was the albedo conversion not applied to the spatial averages? The same bias mentioned in line 77 for the temporal average could be introduced in the temporal averages if RSW instantaneous observations are systematically missing at some parts of the 1x1 degree pixel, due to the change of RSW with latitude.*

Answer: We have amended the sentence to explicitly state that the conversion *is* done before both spatial and temporal averaging (**line 102**).

*5)-(missing) methods section. The current version of the manuscript presents a sequential structure that mixes methodology paragraphs with results paragraphs. For instance, the averaging and gap-filling processes are currently described in two different sections (section 2.1 and section 2.4), while sections 2.2 and 2.3 contain methods and results regarding the impact of*

*missing data. I consider that the readability of the manuscript could improve by splitting methods and results. The methods section could be further split into "dataset production" (a kind of ATBD) and "dataset evaluation" (e.g., the impact of missing data before and after gap-filling). A specific section on the "dataset production" containing all the methodological steps (including an extended version of Fig 1 diagram, which currently only focuses on the averaging process) would be highly valuable for potential users.*

Answer: We have rearranged and reworked these sections in accordance with the suggestion. We now describe the production of an unfilled and filled dataset (both are released Obs4MIPs GERB products) in **section 2** which can be considered to be a 'methods' section and then present the evaluation of each in **section 3** which can be considered a 'results' section.

*6)-I would also suggest adding a specific section or sub-section listing all the attributes of the final product (e.g., spatial and temporal resolution, spatial and temporal coverage, data format, data layers available in the final product, etc.).*

We have added a table (**table 3**) with such information in the data availability section and also added a summary in the conclusions (**lines 557 to 564**).

*7)-Dataset evaluation: The manuscript would significantly improve if the authors included a validation of their dataset against an external TOA satellite-based product. The obvious choice would be using CERES products. This will not only allow benchmarking the new dataset against state-of-the-art products but also having an independent reference to quantify the improvement obtained with some of the methodological steps proposed by the authors (gap-filling, GERB/GERB-like ratios)*

Answer: The original GERB data has already been compared with the CERES products for observationally matched points (i.e. GERB points matched to direct CERES observations). This is a fundamental comparison and evaluation of the GERB observational dataset against the CERES observational dataset (Clerbaux et al 2009, doi:10.106/j.rse.2008.08.016, Parfitt et a 2016, doi: 10.1016/j.rse.2016.09.005, 2016). The diurnal resolution of the CERES SYN is not provided by CERES observations and has been developed and evaluated using observations from GERB (**lines 43 to 47**) (Doelling, et al. 2013 doi: 10.1175/JTECH-D-12-00136.1 & 2016 doi: 10.1175/JTECH-D-15-0147.1) making it neither independent nor the ideal product in respect of its diurnal resolution. All these references have now been included in the paper (**lines 46, 245, 524**). These, much more direct, comparisons exploit the strengths of both datasets: CERES for baseline accuracy, GERB for a correct (observed) representation of the diurnal variation in outgoing fluxes. We argue that comparing the two hourly monthly mean products would essentially mimic these studies but with the complication of combining accuracy and sampling differences that have already been assessed. Hence we argue against performing this exercise. The evaluation presented in this paper deals specifically with the additional problem of creating an average where there is incomplete data which is the only additional consideration in evaluating the fidelity of the GERB Obs4MIPs monthly hourly averages.

**Minor comments**

8) -Line 84: "Hence, twilight and night-time RSW HR fluxes are not included in the averaging to the daily hourly scale if the central solar zenith angle is less than 85 but are used to replace grid-box values when the central solar zenith angle is equal to or exceeds 85" Could you clarify this sentence? Regarding the first part of the sentence, does it mean that 9km pixels with SZA > 85 are not used in the spatial average if the SZA at the center of the 1x1 pixel is less than 85 degrees?

Answer: Correct, if the SZA at the centre of the hour for that 1 degree region is less than 85 degrees the flux value of any GERB HR pixels with SZA > 85 are not included in the spatial or temporal averaging. This is because these pixels are not observations but just a fixed non location dependent twilight model flux used to avoid bias when diurnally averaging and would not be appropriate to include except for a twilight location. By excluding these twilight model values completely when the central point has a SZA < 85 and using only these values when the SZA ≥ 85 we preserve the diurnal bias correction and avoid the model values degrading the observed

values. We have stated more clearly that they are excluded from the spatial and temporal averaging and noted that they are not observed quantities (**lines 112 to 118**).

*9)-Line 194: "empirical narrowband to broadband conversion" Please, include either the equation with the coefficients or a reference to a document describing this conversion.*
Answer: References have been added as requested (**line 162**).

*10)-Figures 2 & 5: Is there any reason to use these unevenly spaced categories (0, 1-5, 6-22, >22) for the number of missing days? If so, explain it. Otherwise, I would suggest using an evenly-spaced color palette or a continuous color palette*
Answer: The colour coding is related to the data included in the filled and unfilled products. We have added information regarding this to the figure caption (**lines 149 to 150**), and referred to later (**lines 312 to 313 and 343**).

*11)-Figures 3, 4, 6, 7, 8, 9: The panel number is not seen very well. Please, take it out of the panel and increase the font size (and/or use bold text).*
Answer: Panel numbers have been moved to the outside of the panels and font size increased as suggested. **Figures 4, 5, 6, 8, 9 and 10.**

*12)-Figure 3: I would suggest using a diverging color palette centered around 0. Otherwise, it is difficult to interpret the differences. I would also suggest describing the four realizations in the figure caption.*
Answer: A diverging colour scale was in the plot but the wrong key was included this has been corrected. **Figure 8**

*13)-Figure 4 & 8. Add a horizontal line (in the background) to better interpret the bias plots.*
Answer:**.** A horizontal zero line has been added to the error distribution mean panels as suggested **Figure 6a,b figure 9c,d and figure 10c,d (for consistency)**

*14)-Figure 6 & 7. As for Figure 3, use a diverging color palette centered around 1*
Answer: We chose a linear colour scale for these plots because it is the spatial and temporal variation in the ratio rather than its difference from 1 that is most significant, this is now clarified (**lines 177 to 179**). A fixed offset would be a simple adjustment but the variability of the difference in time and space requires more complex treatment. A diverging colour scale centred around 1, although sensible in terms of 1 being an important value, would not be useful given the distribution of the ratios. For the RSW less than 1% of the points have a ratio less than 1, and for the OLR less than 2.5% of the points have a ratio greater than 1. Thus, using a colour scale centred around 1 for these plots would limit the plotted colours to around half of those available and would barely make use of the diverging nature of the scale. Whilst a diverging colour scale centred around the mean bias (1.076 for the SW and 0.99 for the OLR) could be used, there is nothing particularly special about the mean, either physically, or in relation to scene or angle. Thus, we feel this choice gives unwanted emphasis to the contrast between points above and below the mean even when they are quite close in value, at the expense of emphasizing the overall range of values which is more relevant to consider. For these reasons we feel a diverging colour scale centred around 1 or the mean would be counterproductive to what these plots are trying to explore.

*15)-Section 2.5: Could you clarify if GERB-like fluxes are used (a) only to replace fully missing 1x1 degree averages, or (b) also to replace missing 9km GERB HR observations before the spatial averaging?*
The corrected GERB-like fluxes are only used to replace fully missing 1x1 degree averages. This has now been more clearly explained (**lines 235 to 239**) and a schematic added to clarify (**Fig. 7**)

*16)-Discuss the challenges to extend this methodology to GERB instruments onboard other MSG satellites, and if you have any plans to undertake this project in the near future.*
A discussion of this has been added to the conclusions as suggested. (**Lines 564 to 569**)

**Summary of changes by section**

A section by section summary of the changes made to the paper in response to the reviews is given below. Changes are linked to reviewer comments using the abbreviations R1 and R2 for reviewer 1 and reviewer 2 respectively followed by the relevant comment number as numbered in the detailed replies below. For example, the title change in response to the comment number 1 from reiviwer 1 is references ad R1.1. All figure, table, section and line numbers in this summary of changes section relate to the revised manuscript.

**Title**: Changed in response to R1.1 (lines 1-2)
**Abstract**:
Changes in response to R1.2 (line 8), R1.3 (line 11), R1.4 (line 20), R1.5 (line 21), R1.6 (line 29), R1.19 (throughout). Additional clarification of the version of the GERB Obs4MIPs products discussed as 'V1.1 filled', in recognition of the alterations in the layout of the paper in response to R2.5. Order of the OLR and RSW swapped for consistency with the order of the dataset references. Changed 'obs4MIPs to Obs4MIPs for consistency throughout the paper.
**Introduction**:
Added context in response to R2.1 (lines 26 to 35), includes changes in response to R1.6 (line 29). Additional reference in response to R1.7 (line 50), phrasing changed in response to R1.19 (throughout). obs4MIPs changed to Obs4MIPs for consistency, abbreviation CF expanded on first use and version specified.
**Section 2.**
Changes to layout and order in response to R2.5 , separating the original section 2 into two distinct sections addressing methodology (new section 2) and evaluation (new section 3). This also addresses concerns expressed in R1.21 and R1.22, goes some way to address confusion noted in R1.29 and addresses the balance of the paper noted in the overarching comment made by R1. An additional introductory paragraph has been included to support the change in layout and explain the two versions of the GERB Obs4MIPs products discussed. The original section 2.3 'Evaluation of the impact of missing data' has been been moved to section 3.1 and becomes 'Impact of missing data on the fidelity of the unfilled GERB Obs4MIPs products'. Whilst the original section 2.5 'Evaluation of filled data record' has been moved to section 3.2 and becomes 'Fidelity of the filled GERB Obs4MIPs products'.
Alterations and additional explanations in response to R1.8 & R2.2 (lines 77 to 82), R1.9 (line 80), R1.10 (line 96), R1.11 (lines 98 to 1010),R2.4, R1.12 & R2.3 (lines 104 to 108), R2.8 (lines 112 to 118), R1.14 (line 155), R2.9 (line 162) R1.15 (lines 166 to 169), R1.16 (lines 170 to 171), R2.14 (lines 177 to 179), R1.18 (lines 210 to 212), R1.19 (throughout), R1.20 and R1.21 (lines 216 to 224), R2.15 (lines 235 to 239). Changes to figures numbered 4, 5 and 6 in the updated manuscript in response to R1.17, R2.9, R1.20, R2.11, R2.12, R2.13. Additional text in caption to figure 2 in response to R2.10. An additional figure 7 added to support the change and address R2.15.
**Section 3**.
Evaluation presented separately from the methodology in response to R2.5 with an added introductory paragraph to the section to support the new layout. Comprises the text originally presented in section 2.3 and 2.5 as the new section 3.1 and 3.2. Additions

referencing the increase in amount of data available in the unfilled and filled products with reference to figure 2 (lines 310 to 313 and lines 342 to 345) as part of the changes in response to R2.5 to discuss and compare V 1.0 and V 1.1 products and with consideration of R2.10. Changes to figures 8, 9 and 10 in response to R1.13, R1.22, R2.11, R2.12, R2.13.

**Section 4**

AMIP defined and referenced in response to R1.23 (line 356). Section 4.1 has been significantly simplified in response to R1.24 (366 to 372). The scope of the initial spatial comparison is clarified (lines 358 to 359 and 400 to 401) and alterations made to figure 11 in response to R1.25a. Updates to figures 14 and 15 have been made in response to R1.25d. Clarification of the GERB Obs4MIPs product being used as V1.1 to reflect the possible ambiguity introduced by the changes to section 2. Changed obs4MIPs to Obs4MIPs for consistency through the paper. Addition and reference included as suggested in R1.26 (lines 396 to 397) with clarification 'at least in part' added for consistency with this addition regarding scale.

**Data availability**:

Added table 3 on data characteristics in response to R2.6. Reference to V1.0 and V1.1 Obs4MIPs products now made to reflect the changes in section 2.

**Conclusion**

Changes to the first paragraph for compatibility with the presentation changes in section 2 and 3. Additional reference to previous GERB comparisons and validation is included in response R2.7 (line 524). Added sentence in response to R1.29 (lines 530 to 536). Additions regarding the released data and plans for other GERB instruments in response to R2.6 and R2.16 (lines 557 to 569), these include specifics regarding uncertainty in response to R1.30 (lines 562 to 564)

**Author contributions:**

Addition to reflect work carried out in response to reviews.

**Acknowledgements:**

Added in response to R1.32 and to acknowledge pertinent funding sources.

**References:**

Additions to support changes and additions in text.